# CAUSAL CONFUSION AND REWARD MISIDENTIFICATION IN PREFERENCE-BASED REWARD LEARNING

**Jeremy Tien**
University of California, Berkeley
jtien@berkeley.edu

**Jerry Zhi-Yang He**
University of California, Berkeley

**Zackory Erickson**
Carnegie Mellon University

**Anca D. Dragan**
University of California, Berkeley

**Daniel S. Brown**
University of Utah

## ABSTRACT

Learning policies via preference-based reward learning is an increasingly popular method for customizing agent behavior, but has been shown anecdotally to be prone to spurious correlations and reward hacking behaviors. While much prior work focuses on causal confusion in reinforcement learning and behavioral cloning, we focus on a systematic study of causal confusion and reward misidentification when learning from preferences. In particular, we perform a series of sensitivity and ablation analyses on several benchmark domains where rewards learned from preferences achieve minimal test error but fail to generalize to out-of-distribution states—resulting in poor policy performance when optimized. We find that the presence of non-causal distractor features, noise in the stated preferences, and partial state observability can all exacerbate reward misidentification. We also identify a set of methods with which to interpret misidentified learned rewards. In general, we observe that optimizing misidentified rewards drives the policy off the reward's training distribution, resulting in high predicted (learned) rewards but low true rewards. These findings illuminate the susceptibility of preference learning to reward misidentification and causal confusion—failure to consider even one of many factors can result in unexpected, undesirable behavior.

## 1 INTRODUCTION

Preference-based reward learning (Wirth et al., 2017; Christiano et al., 2017; Stiennon et al., 2020; Shin et al., 2023) is a popular technique for adapting AI systems to individual preferences and learning specifications for tasks without requiring demonstrations or an explicit reward function. However, anecdotal evidence suggests that these methods are prone to learning rewards that pick up on spurious correlations in the data and miss the true underlying causal structure, especially when learning from limited numbers of preferences (Christiano et al., 2017; Ibarz et al., 2018; Javed et al., 2021). While the effects of reward misspecification have recently been studied in the context of reinforcement learning agents that optimize a proxy reward function (Pan et al., 2022), and the effects of causal confusion have been emphasized in behavioral cloning approaches that directly mimic an expert (De Haan et al., 2019; Zhang et al., 2020; Swamy et al., 2022), we provide the first systematic study of reward misidentification and causal confusion when learning reward functions.

Consider the assistive feeding task in Fig. 2b. Note that all successful robot executions will move the spoon toward the mouth in the area in front of the patient's face—few, if any, executions demonstrate behavior behind the patient's head (the exact likelihood depending on how trajectories for preference queries are generated). In practice, we find that the learned reward will often pick up on the signed difference between the spoon and the mouth rather than the absolute value of the distance. The two correlate on the training data, but using the former instead of the latter leads to the robot thinking that moving behind the patient's head is even better than feeding the person!

Our contribution is a study of causal confusion and reward misidentification as it occurs in preference-based reward learning. First, we demonstrate the failure of preference-based reward learning to produce causal rewards that lead to desirable behavior on three benchmark tasks, even

when given large amounts of preference data—in these settings, the learned reward has high test accuracy but leads to poor policies when optimized. We then study the effect of several factors related to reward model specification and training: the presence of non-causal distractor features, reward model capacity, noise in the stated preferences, and partial state observability. For each of these, we perform an analysis of what errors the learned reward has, how it compares to the ground truth reward, and the amount of distribution shift it induces during policy optimization. Another of our contributions is to point to the importance of data coverage and interactive techniques that iteratively or actively query for feedback to help learners disambiguate causal features from correlates. Overall, our findings caution that there are many aspects that can make preference learning challenging, from the way we define state—what to include and what *not* to include in the input to the learned model—to how we mitigate the effects of noisy or biased preference data.

## 1.1 RELATED WORK

**Reward Hacking** and gaming behaviors are known to commonly occur in reinforcement learning (Ng et al., 1999; Krakovna et al., 2020) and reward learning (Christiano et al., 2017; Ibarz et al., 2018; He & Dragan, 2021). However, these behaviors are often only mentioned anecdotally (Krakovna et al., 2020). Recently, Pan et al. (2022) proposed to systematically analyze reward misspecification in RL by creating a set of domains where the agent optimizes a *hand-engineered* proxy reward function and then studying when this leads to incorrect behavior. By contrast, we study cases where the reward function *must be learned*.

**Preference Learning** is a popular method for training AI systems when a reward function is unavailable (Wirth et al., 2017; Christiano et al., 2017; Biyik & Sadigh, 2018; Brown et al., 2020a; Ouyang et al., 2022; Shin et al., 2023; Liu et al., 2023). Preferences are often easier to provide than demonstrations or raw numerical values (Wirth et al., 2017) since they do not require expert proficiency or fine-grained feedback. However, optimizing a reward function that has been trained using preferences can sometimes lead to unintended behaviors. Anecdotal evidence of this has been documented for Atari games (Christiano et al., 2017; Ibarz et al., 2018; Brown et al., 2020a), simple robot navigation tasks (Javed et al., 2021), and language model fine-tuning (Stiennon et al., 2020); however, there has been no systematic study of causal confusion when learning reward functions.

**Causal Confusion in Imitation Learning** has previously been studied in the context of behavioral cloning (Pomerleau, 1988; Torabi et al., 2018). Prior work shows that behavioral cloning approaches suffer causal confusion due to "causal misidentification," where giving imitation learning policies more information leads to worse performance (De Haan et al., 2019) due to temporally correlated noise in expert actions (Swamy et al., 2022). Similarly, we find strong evidence of causal misidentification when expert noise is present. Zhang et al. (2020) use causal diagrams to investigate causal confusion for simplified imitation learning tasks with discrete actions and small numbers of states when the features available to the demonstrator are different from those of the imitator. By contrast, we study the effects of changes to the observation space when performing preference learning over continuous states and actions and when using non-linear reward function approximation.

**Goal Misgeneralization** happens when an RL agent has a known goal and some environment features are correlated and predictive of the reward on the training distribution but not out of distribution (Langosco et al., 2022; Shah et al., 2022). This setting is similar to ours in that there is misidentification of the state features that are causal with respect to the reward. However, goal misgeneralization assumes that the ground-truth reward signal is always present during training. We show that when learning a reward function, the learned reward can be misidentified, leading to poor RL performance. By contrast, Langosco et al. (2022) and Shah et al. (2022) show that even if the learned reward is perfect, RL can still fail due to spurious correlations during training.

## 1.2 BACKGROUND: REWARD LEARNING FROM PREFERENCES

We model the environment as a finite horizon MDP (Puterman, 2014), with state space $\mathcal{S}$, action space $\mathcal{A}$, horizon $T$, and reward function $r : \mathcal{S} \times \mathcal{A} \to \mathbb{R}$. The reward function is unobserved and must be learned from preferences over trajectories. Using the Bradley-Terry model (Bradley & Terry, 1952), the probability a trajectory $\tau_B$ is preferred over another trajectory $\tau_A$ is given by

$$P(\tau_A \prec \tau_B) = \frac{\exp(r(\tau_B))}{\exp(r(\tau_A)) + \exp(r(\tau_B))}, \tag{1}$$

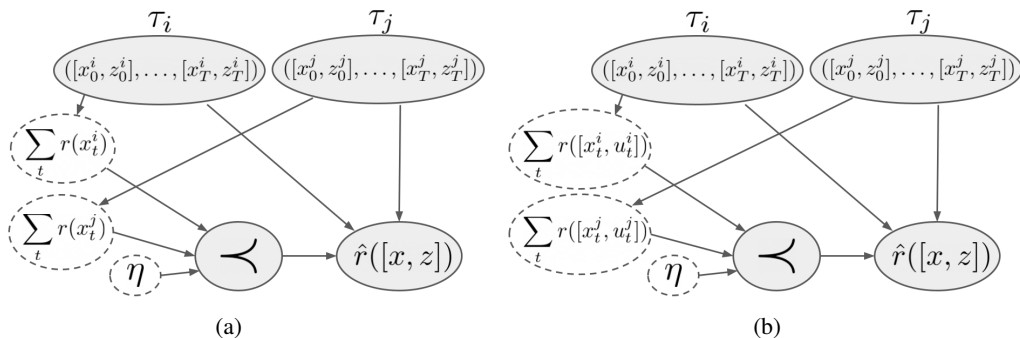

Figure 1: **Causal structure of preference-based reward learning.** The reward function $\hat{r}$ is learned from preference labels over trajectory pairs $(\tau_i, \tau_j)$. Unobserved variables are denoted by dashed lines. Unobservable user noise $\eta$ affects the preference node. In (a) the true reward is not affected by nuisance variables $z$. In (b) the true reward is based on unobserved state features $u$.

where $r(\tau) = \sum_{(s,a) \in \tau} r(s, a)$ and $\tau = (s_0, a_0, \dots, s_T, a_T)$.

To learn a reward function from preferences, we assume access to a set of pairwise preference labels $\mathcal{P}$ over trajectories $\tau_1, \dots, \tau_N$, where $(i, j) \in \mathcal{P}$ implies that $\tau_i \prec \tau_j$. We then optimize a reward function $r_\theta : \mathcal{S} \times \mathcal{A} \to \mathbb{R}$ parameterized by $\theta$ that maximizes the following likelihood (see Alg. 1):

$$\mathcal{L}(\theta) = \prod_{(i,j) \in \mathcal{P}} \frac{\exp(r_\theta(\tau_j))}{\exp(r_\theta(\tau_i)) + \exp(r_\theta(\tau_j))}. \qquad (2)$$

## 2 REWARD MISIDENTIFICATION

Fig. 1 displays the causal structure of preference-based reward learning where pairwise preferences (in the form of a binary label) are given based on an observed reward function $r$ (Fig. 1a). There are features $x_t^i$ that are causal and influence $r(x)$, as well as other features $z_t^i$ that are nuisance variables and have no bearing on $r(x)$. Note that $z_t^i$ may very well exhibit correlations with $x_t^i$, potentially due to their sharing of the same causal parent or there being biases during data collection. Given preference labels over trajectory pairs, the goal of preference-based reward learning is to learn a reward function $\hat{r}(x, z)$ that best matches the stated preferences. In Fig. 1b, there are state features $u$ which are causal with respect to the true reward, but are unobserved by the learning agent. In both cases, unobserved human bias and noise, denoted by $\eta$, also affects the preferences labels.

The learned reward $\hat{r}$ may be able to achieve low and even near-perfect performance on a held-out test set by making use of 1) nuisance variables that correlate with causal variables or 2) faulty/incomplete correlations between causal variables and $r$ that happen to hold true for the training data distribution. However, performing reinforcement learning on misidentified learned reward values $\hat{r}([x, z])$ leads to distribution shift, resulting in behaviors with low performance under the true reward function $r$. We define this behavior of learning a reward that achieves low test error but results in poor performance (under the true reward function) when the learned reward function is optimized via RL as *reward misidentification*. We note that the causal graphs we provide in Fig. 1 are meant to shed light on a typical way misidentification can occur. In reality, the sources of misidentification can vary widely—a variable may be causal in a certain context and not in another, causal variables may not be combined properly, etc.

## 3 EXPERIMENTAL SETUP

To facilitate reproducibility and encourage future research on causal reward learning, we open-source our code and training datasets: `https://sites.google.com/view/causal-reward-confusion`. This combination of domains and training data forms the first set of benchmarks for studying reward misidentification and causal reward confusion.

**Environments for Preference Learning.** We identify a set of benchmarks that exhibit reward misidentification. In ***Reacher*** (Brockman et al., 2016) (Fig. 2a), the goal is to move an end effector to a desired goal location. In ***Feeding*** (Erickson et al., 2020) (Fig. 2b), the goal is to feed the human using a spoon carrying pieces of food. Finally, in ***Itch Scratching*** (Erickson et al., 2020) (Fig. 2c), the goal is to scratch an itch location on the human's arm.

**True Rewards and Preference Generation.** Each domain has a predefined "true" reward function $r$ (see Appx. A.5). This enables us to create synthetic demonstrations and preference labels via *noise injection*: adding different amounts of noise to an expert policy trained on $r$ (details in Appx. A.2). As shown by Brown et al. (2020b), adding this type of disturbance will result in monotonically decreasing performance in expectation and produce a diverse dataset for preference learning. Gleave et al. (2020) propose a similar approach that switches between an expert policy and a random policy to produce a good coverage distribution over states. In Appx. A.12 we compare generating trajectories using noise injection versus using different RL checkpoints, as proposed by Brown et al. (2019), and find that noise injection leads to similar or better performance. Note that while we use the ground-truth reward function for obtaining preference labels, we assume no access to this reward function during policy learning; we first learn a model $r_\theta$ of the true reward function from preference labels $\mathcal{P}$, and then use RL on the learned reward

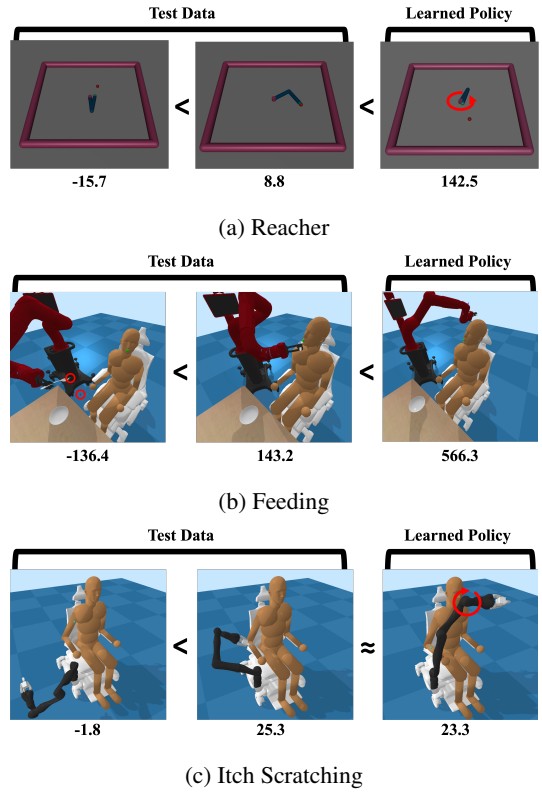

(a) Reacher

(b) Feeding

(c) Itch Scratching

Figure 2: **Poor behaviors resulting from learned rewards** (rightmost column), despite high accuracy on the test data (left two columns). The predicted trajectory rewards produced by the learned reward function are displayed under each image; each image corresponds to a trajectory.

function to produce a policy. We then evaluate the learned policy on the true reward function $r$.

**Evaluating Learned Reward Functions.** To establish that a learned reward misidentifies the causal structure, we first check for ***low test error*** and establish that the learned reward performs well on unseen in-distribution test data (thereby ruling out model selection and training failures like not having enough data, capacity, or regularization). We then show that the learned reward fails in two ways: 1) it leads to a policy (called *PREF*) that has ***poor performance with respect to the true reward***, and 2) it ***prefers its poorly-performing optimized policy PREF over the optimal policy with respect to the true reward (GT)***, indicating that PREF's poor performance is not due to RL failures. Finally, we analyze the learned reward qualitatively via gradient saliency maps and quantitatively via the EPIC pseudometric (Gleave et al., 2020) and KL divergence to elucidate the effects of the reward error by quantifying the distribution shift induced by policy optimization of the learned reward. Details on hyperparameters and evaluation methods are in Appx. A.1 and A.6.

## 4 EVIDENCE OF CAUSAL CONFUSION

Before varying different factors that affect the performance of the learned reward, we start with a generous setting where we provide large amounts of data and add features to the default observation such that all necessary information needed to infer the ground truth reward, TRUE, is available. We produce the preference training data as detailed in the previous section. Table 1 details the results.

We find that the learned reward achieves high preference test accuracies that are comparable to the training accuracy. This indicates that the learned model does not overfit, and there is sufficient

Table 1: **Empirical evidence of causal confusion.** We compare policies optimized with a reward learned from preferences (PREF) against policies optimized with the true reward (GT). State features on which preferences are based are fully-observable. Reward functions were trained with 52326 unique pairwise preferences. Both PREF and GT are optimized with 1M RL iterations and averaged over 3 seeds. Despite high pairwise preference classification test accuracy, the policy performance achieved by PREF under the true reward is very low compared with GT. However, *the reward learned from preferences consistently prefers PREF over GT.* This suggests that preference-based reward learning fails to learn a good reward for each of these tasks.

| | PREF. LEARNING ACC. | | | RL POLICY PERFORMANCE | | |
| DOMAIN | TRAIN | VAL | TEST | LEARNED (PREF/GT) | TRUE (PREF/GT) | SUCCESS (PREF/GT) |
| --- | --- | --- | --- | --- | --- | --- |
| REACHER | 0.954 | 0.956 | **0.966** | **44.988** / 3.395 | -42.716 / **-5.560** | 0.100 / **0.827** |
| FEEDING | 0.987 | 0.976 | **0.976** | **277.152** / 124.016 | -27.432 / **128.933** | 0.603 / **0.990** |
| ITCHING | 0.954 | 0.933 | **0.928** | **16.588** / 10.282 | -47.190 / **248.397** | 0.013 / **0.970** |

model capacity and data. In later sections, we observe that even models with over 99% test accuracy sometimes fail to produce good polices. We also find that the learned reward *prefers* PREF to GT. This shows that the poor performance is not due to an issue in RL training. Unfortunately, the actual performance of PREF is disastrous: it has poor TRUE reward (compared to GT) and poor success rates. Overall, the learned reward incentivizes poor behavior, despite high test accuracy.

For Reacher, we observe that PREF chooses to simply spin very fast rather than reaching for the target. We note, however, that the learned reward correctly classifies the leftmost image in Fig. 2a (where the agent just folds its arm) as being worse than the middle image (where the agent successfully reaches the target), and does so with 96.6% accuracy on such diverse pairs of comparisons. One would think that this would apply to the behavior in the rightmost image, where the agent folds its arm as well (and subsequently spins), but the learned reward turns out to strongly prefer the rightmost case. For Feeding, the learned reward encourages minimizing the *signed* difference between the spoon and the mouth rather than the *absolute value* of the difference. This is because the majority of trajectories approach the mouth from in front. Fig. 8b shows that there are far more states that have low reward at negative 'diff_y' values than states that have low reward at positive 'diff_y' values. As a result, the learned reward (Fig. 9b) correctly identifies spilling food (left) as being worse than feeding (middle), but goes further and incentivizes bringing the spoon towards and even *behind* the head in an attempt to minimize the signed difference. For Itch Scratching, the learned reward correctly identifies flailing (left) as being worse than actual scratching (middle). However, as seen in Fig. 9c, the Itch Scratching agent (spuriously) learns a higher weight on two components of the action (corresponding to two robot arm joints), which results in the agent turning the last section of the arm in a circle while trying to keep the end effector close to the itch target—another type of flailing! It also falls into the same trap as the Feeding agent in minimizing the signed rather than the absolute difference (Fig. 8c shows there is a bias toward negative values of 'diff_x' rather than an even distribution of states across both positive and negative values). Fig. 2 summarizes this behavior. Appx. A.3 provides plots of the aforementioned features' spurious correlations with the true reward and gradient saliency maps of the learned rewards.

## 5  FACTORS THAT MAY LEAD TO CAUSAL REWARD CONFUSION

We examine various factors of the preference-based reward learning problem setup and their effects on reward misidentification. The motivation for exploring each is as follows: Our experiments with **Distractor Features** draw directly from the findings in Sec. 4, where we find that certain non-causal features may be spuriously correlated with the reward. Experiments on **Model Capacity** are inspired by Pan et al. (2022)'s findings on the effects of increased agent capabilities on reward hacking. Exploring **Noise in Stated Preferences** is inspired by the fact that preference data collected from humans is often rife with various biases and noise. Studying **Partial Observability of Causal Features** is motivated by the fact that it is not always possible to fully-observe all causal features in the real world. **Complex Causal Features** is inspired by the observation that the causal reward is

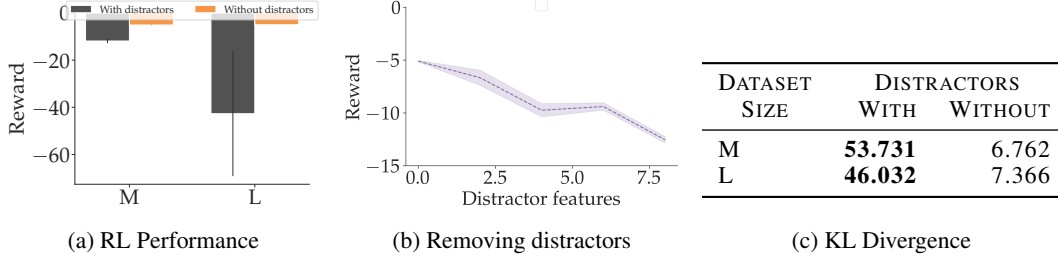

(a) RL Performance      (b) Removing distractors      (c) KL Divergence

Figure 3: **Distractor features.** Fig. 3a compares performance with distractors present in the learned reward with performance without distractors present in the learned reward across two dataset sizes. Fig. 3b is a sensitivity analysis on the number of distractor features done on the M dataset size case from Fig. 3a. In Fig. 3c, we see that distractor features result in a much larger distribution shift (as measured by the KL divergence between state-action pair distributions at reward learning and RL).

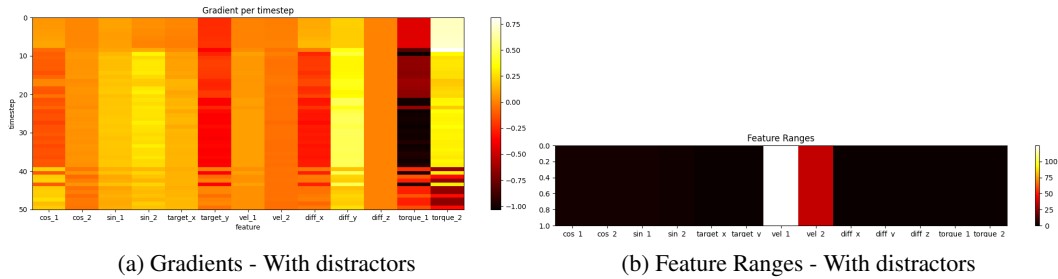

(a) Gradients - With distractors      (b) Feature Ranges - With distractors

Figure 4: **Saliency maps: distractor features.** Fig. 4a is a gradient saliency map of the (misidentified) learned reward. Fig. 4b is a heatmap displaying the ranges of each feature over the course of a trajectory produced by the learned reward. The large range of the 'vel_1' feature and its corresponding positive gradient are the reason why the trajectory does well under the learned reward.

often a complex function of state variables and, in complex tasks, features may be causal in some contexts and non-causal in others. The above factors (which may often be overlooked) can contribute significantly to reward misidentification and the eventual success or failure of the reward model.

## 5.1 DISTRACTOR FEATURES

One reason for reward misidentification is the presence of nuisance features in the input that are spuriously correlated with preference labels. To study this effect, we incrementally remove such 'distractor' features and test the impact this has on the learned reward. For example, for Reacher, we know that joint angles and angular velocities are not causal to the ground truth reward, which is based purely on distance between end effector and target and the norm of the action (a control penalty) (see Appx. A.4 for a complete list of causal and distractor features). As we see in Fig. 3, removing all distractors drastically improves performance, and, indeed, the more such features are left in the input, the worse the performance gets in Reacher, almost linearly. Granted, removing distractors does also slightly improve the validation and test performance in this case—indicating that there is some signal in the training data to help discern the spuriousness of certain features. Nonetheless, over the several experiments in this paper, we find that the validation error is not strongly correlated to performance—in some cases, increasing the validation accuracy results in worse performance.

Fig. 2a depicts the policy of the learned reward with distractors—the Reacher robot learns to simply fold its arm and spin. Looking at the gradient saliency maps (Fig. 4) illuminates why this is the case. Firstly, in Fig. 4a, we observe that the learned reward is misaligned with respect to 'diff_y', the feature corresponding to the difference between end effector position and target position along the y-axis; specifically, we note that the learned reward actually rewards an increase in 'diff_y', rather than a decrease. Next, we notice that the reward doesn't penalize action evenly. It penalizes 'torque_1' (the one responsible for the spinning), but rewards 'torque_2' across nearly all timesteps. It should instead be largely a negative penalty, as seen in Fig. 11a. Lastly and perhaps most importantly, we observe in Fig. 4b that 'vel_1' and 'vel_2' have very large feature ranges, corresponding to the large

variations in angular velocity achieved by the Reacher robot's spinning behavior. Looking back to the gradient for each of these velocity features in Fig. 4a, we observe that the reward has a slight positive gradient with respect to each which stems from a slight correlation between angular velocity and reward. Fig. 8a shows this correlation; there is a slight bias in the training data toward states with high reward and high velocity. By spinning fast, the Reacher robot is thus able to achieve much higher (learned) reward than performing the proper reaching action. The KL divergence (Table 3c) between the distribution of observation-action pairs seen during reward learning and those seen during policy optimization provides further insights—in incentivizing the Reacher robot to spin fast, it leads the RL optimization toward states that were not seen during reward learning (specifically, states where the robot is spinning very fast).

Appx. A.7 displays results for the other tasks. Similar to the Reacher task, we find that removing distractor features generally helps performance. Note that the reason removing non-causal distractor features does not appear beneficial for Feeding is that the main spurious correlation (discussed in Section 4) involves one of the *causal* features—namely, the difference between spoon position and target position. Thus, removing purely non-causal features fails to address this issue in Feeding.

## 5.2 MODEL CAPACITY

In Appx. A.8, we study the effect of model capacity on the learned reward. We find that, despite careful tuning of hyperparameters with each model and dataset size, increasing the capacity of the reward model does not necessarily result in an increase in subsequent policy performance.

## 5.3 NOISE IN STATED PREFERENCES

Because user noise is often inevitable, we explore the effect of various types of user noise on reward misidentification. Following Lee et al. (2021), we modify Eq. 1 to include a rationality constant $\beta$ and a myopic discount factor $\gamma \in (0, 1]$:

$$P(\tau_A \prec \tau_B) = \frac{\exp(\beta \sum_{t=1}^{H} \gamma^{H-t} r(s_t^B, a_t^B))}{\exp(\beta \sum_{t=1}^{H} \gamma^{H-t} r(s_t^A, a_t^A)) + \exp(\beta \sum_{t=1}^{H} \gamma^{H-t} r(s_t^B, a_t^B))}. \tag{3}$$

Using Eq. 3, we explore four types of user noise (varied independently of each other): STOCHASTIC, where the user is rational with $\beta = 1$; MYOPIC, where earlier rewards are discounted with a $\gamma = 0.99$; SKIP, where the user skips a pair of trajectories if both have rewards below a certain threshold; and MISTAKES, where the preference label is randomly flipped with probability $\epsilon = 0.1$. ORACLE has $\beta = \infty$. Results are displayed in Fig. 5.

We find that noise in the stated preferences exacerbates reward misidentification—test accuracy stays high while policy performance plunges (Fig. 5a). Notably, two instances of user noise, Stochastic and Skip, have test accuracies greater than or equal to the Oracle (despite far worse performance). Importantly, the poor performance of the learned policies is not well predicted by the validation accuracies, with models achieving 0.995 resulting in worse alignment than models trained on less data with lower validation accuracy of 0.985. Additional results are located in Appx. A.9.

We use EPIC (Gleave et al., 2020) as one way to measure the difference betweeen reward functions. Table 9 shows that when the coverage distribution for EPIC is chosen as the distribution of state-action pairs seen during reward learning, the misidentified rewards due to user noise are closer to the ground truth rewards, in EPIC distance, than rewards learned without user noise. However, the opposite is the case when the coverage distribution is chosen to be the distribution of state-action pairs seen during RL. This implies that the misidentified reward model "mimics" the ground truth reward on the reward learning distribution but fails to generalize when taken out of distribution by RL. We further observe in Table 8 that the KL divergence between reward learning and reinforcement learning state distributions is greater when the learned reward contains noise (Stochastic). This indicates that the rewards trained with noisy data are misidentified: they have low test error but incentivize RL to deviate from the optimal behavior encountered during reward learning.

## 5.4 PARTIAL OBSERVABILITY OF CAUSAL FEATURES

Partial observability over aspects of the state on which the user's preferences are based is nearly inevitable in the real world. Interestingly, RL with the ground truth reward is able to learn proper

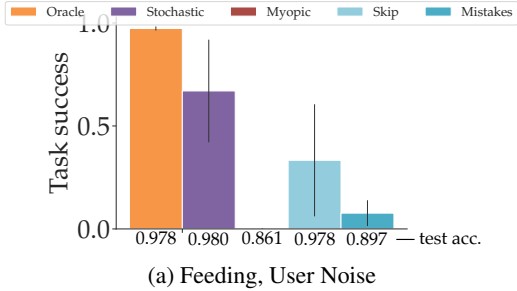

(a) Feeding, User Noise

Figure 5: **Noise in stated preferences** results in significantly degraded performance.

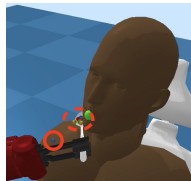

| OBSERVABILITY | PERFORMANCE | |
| --- | --- | --- |
| | REWARD | SUCCESS |
| PREF-FULL | 126.120 | 0.973 |
| PREF-PART | -120.825 | 0.040 |
| GT-PART | 128.933 | 0.990 |

Figure 6: **Partial observability.** Minimizing spoon-mouth distance without observability of food leads to spilling food (left). PREF and GT refer to policies optimized with the learned and true reward, respectively. FULL and PART refer to the amount of observability over causal features.

feeding behavior *in spite of this lack of critical state information*. However, as displayed in Fig. 6, reward learning is only successful when the model is able to observe all the causal reward features.

In Appx. A.10, we analyze gradient saliency plots and find that partial observability causes the reward model to incorrectly over-weigh causal features that are available and pick up on spurious correlations with non-causal variables. With Feeding, we observe that the reward model learns stronger weights on one of the joint angles and two components of the action, all of which have no bearing on the true reward. Simultaneously, the reward also learns a greater weight on the second component of the feature corresponding to the vector distance between the end effector and the mouth. This results in the behavior depicted in Fig. 6—the robot manipulator is able to successfully maneuver the spoon close to the patient's mouth (by observing the distance feature), but does so without ensuring that the food particles themselves stay on the spoon and end up in the patient's mouth.

## 5.5 COMPLEX CAUSAL FEATURES

To explore how results may differ as we increase the complexity of the task, we evaluate reward misidentification on the complex task of Itch Scratching. We find that even after increasing the amount of training data for reward learning (Fig. 7a), increasing the reward model capacity (Fig. 7b), removing distractor features that are not causally related to the ground truth reward, having perfectly stated preferences, and ensuring full observability over all the features that are involved in the ground truth reward, performing preference-based reward learning still fails to produce a policy that successfully scratches the itch location on the patient. However, as shown in Table 1, a standard model-free RL algorithm trained using the ground truth reward is able to solve such a task.

The task's scratching motion requires not only making contact with the target using the end-effector, but also that the target contact position be greater than a $\delta$ away from the previous target contact position and that the exerted force be no more than a $F_{max}$. Although the preferences are based on how well this motion is performed and despite the reward model having access to all the necessary aforementioned information (including information about the state at the previous timestep; see Appx. A.4), we find that the reward model is not able to learn the scratching motion. As seen in Fig. 7c, once we explicitly include a high-level indicator feature, "scratched" (whether the robot has successfully performed the scratching motion), performance drastically increases. We suspect that the reward model's tendency to pick up on spurious correlations that occur consistently over the course of the trajectory involving just a few variables prevents it from learning the true causal

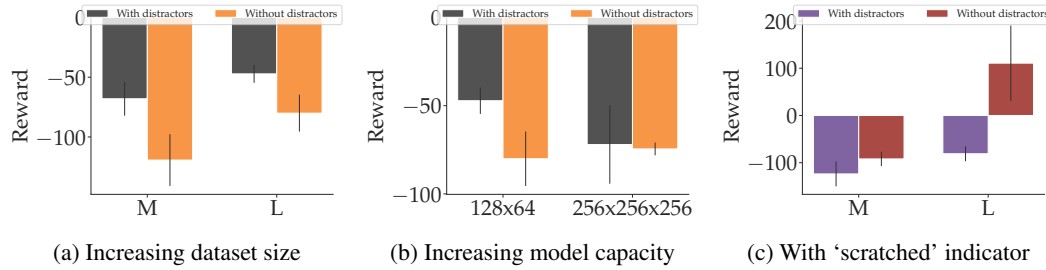

Figure 7: **Complex causal features.** Learning the "scratched" feature from low-level causal features is difficult, despite increasing dataset size and model capacity. Performance significantly improves when given access to a high-level 'scratched' feature.

relation that involves many variables, each of which are causal only in a particular context. In Appx. A.11 we analyze this further and find that the learned reward without an explicit "scratched" feature leads to a greater amount of distribution shift. Future work should address the problem of learning this kind of complex, multi-feature causal relationship.

## 6 CONCLUSION

Our work provides an analysis of reward misidentification and causal reward confusion. We identify three tasks where preference learning often results in misaligned learned reward models. Interestingly, these models have good validation and test accuracy—sometimes even 99.5%—and seem to distinguish the basics of task success versus failure. However, optimizing these rewards via RL pushes the policy outside of the training distribution, where the model falsely believes the reward is higher—we demonstrate this via saliency maps, EPIC distance, and KL divergence and by examining the resulting behaviors. This out-of-distribution effect results in policies that achieve high learned rewards but have poor true rewards. We find that it is easy for reward models to pick up on non-causal features, as some issues go away when we eliminate these non-causal features from the input. Furthermore, noisy preference data aggravates poor generalization. And when not all causal features are observable, the learned reward model will struggle even though there exists a high-performing policy that only uses observable state information.

Based on our results, we have identified several directions for future work. First, our results show that reward misidentification induces a distribution shift such that the learned reward appears deceptively close to the true reward on the training distribution, despite leading to misaligned behavior when optimized via RL. As such, future work should investigate methods for penalizing excess exploration beyond the training data distribution. In Appx. A.13, we examine the effect of adding a direct penalty on the KL divergence into the cost function during RL. We find that the RL agent is again able to hack the learned reward—performance becomes even worse because of the additional degree of freedom in the reward function afforded by the discriminator penalty. However, we hypothesize there are other ways to successfully incorporate such a penalty. Next, our results demonstrate that spurious ("nuisance") features can significantly increase the chance of reward misidentification. Thus, in cases where we can query for human knowledge on exactly which features are spurious or causal, we can use this feedback to learn which features are non-causal or remove non-causal features entirely. Further, our results indicate that high-level features such as the "scratch" feature are difficult for neural networks to learn, even when all the necessary low-level information is available. Future work should examine incorporating methods for learning high-level features (Bobu et al., 2021). Using alternatives to EPIC (Gleave et al., 2020) such as DARD (Wulfe et al., 2022) to compare learned rewards may also prove fruitful in detecting and alleviating reward misidentification. Finally, we recognize that active and iterative methods for data acquisition is also a widely popular solution—as such, we provide a preliminary exploration in Appx. A.14.

Our work cautions that reward learning is brittle—natural choices for acquiring data, deciding on the amount of data, or defining the input space can lead to models that seem very close to the true reward, but lead to spectacular failures when optimized. While active and iterative methods for data acquisition may alleviate some of these issues, learning reward models when some of the causal features are unobservable remains an open challenge.

ACKNOWLEDGMENTS

The authors would like to thank the InterACT lab for their insightful feedback and fruitful discussions over the course of this work. This work was supported in part by an ONR YIP award.

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

# A    APPENDIX

## A.1    PREFERENCE LEARNING TRAINING DETAILS

To learn a reward function from preferences, we assume access to a set of pairwise preference labels $\mathcal{P}$ over trajectories $\tau_1, \ldots, \tau_N$, where $(i, j) \in \mathcal{P}$ implies that $\tau_i \prec \tau_j$. We then optimize a reward function $r_\theta : \mathcal{S} \times \mathcal{A} \to \mathbb{R}$ parameterized by $\theta$ that maximizes the likelihood:

$$\mathcal{L}(\theta) = \prod_{(i,j) \in \mathcal{P}} \frac{\exp(r_\theta(\tau_j))}{\exp(r_\theta(\tau_i)) + \exp(r_\theta(\tau_j))}. \tag{4}$$

This likelihood function is differentiable, allowing us to leverage non-linear function approximation to learn the reward function from trajectory preferences. In practice, we use the Adam optimizer in PyTorch to learn the reward function, $r_\theta$, and then use PPO Schulman et al. (2017) or SAC Haarnoja et al. (2018) for policy optimization given $r_\theta$.

We perform preference learning on three dataset sizes (given in terms of unique pairwise comparisons): SMALL (780), MEDIUM (7140), and LARGE (52326). Our test set is composed of 1770 unique pairwise preferences drawn from a disjoint set of 60 trajectories. See Appx. A.2 for dataset generation details. Hyperparameters—learning rate and weight decay—are tuned coarsely using the MEDIUM dataset size due to runtime limits and cost of computation. The tuned hyperparameters (best performance on a held-out validation set) for each environment are as follows: Reacher: weightdecay=0.0001, lr=0.01, Feeding: weightdecay=0.00001, lr=0.001, Itch Scratching: weightdecay=0.001, lr=0.001.

Using the dataset of preferences (SMALL, MEDIUM, or LARGE) obtained offline, we train a neural network reward function approximator with two hidden layers (128 units and 64 units, respectively) and Leaky ReLU activations, after which we perform 1,000,000 timesteps of reinforcement learning with PPO Schulman et al. (2017) (for Feeding and Itch Scratching) and SAC Haarnoja et al. (2018) (for Reacher) using the learned reward function in place of the ground-truth reward function. We optimize the reward function approximator using Adam with weight decay and early-stopping on the validation loss (with a patience of 10 epochs). The full preference-based reward learning procedure we use is detailed in Alg. 1. The hyperparameters for the PPO and SAC agents are as follows (if not specified, then hyperparameters are set to RLLib's default values):

PPO:

- training batch size = 19200
- number of SGD iterations = 50
- SGD minibatch size = 128
- lambda = 0.95
- dimensions of fcnet hidden layers = [100, 100]

SAC:

- timesteps_per_iteration = 400
- learning_starts = 1000
- Q_model fcnet_hiddens = [100, 100]
- policy_model fcnet_hiddens = [100, 100]
- train_batch_size = 4096
- actor_learning_rate = 3e-3
- critic_learning_rate = 3e-3
- entropy_learning_rate = 3e-3

---

**Algorithm 1** Preference-Based Reward Learning

---

**Input:** Training dataset of pairwise preferences $d_{train}$

    $\hat{r} \leftarrow$ initialize network (random)

    **for** $epoch \in (0, 100)$ **do**

        **for** $traj_i, traj_j, label \in d_{train}$ **do**

            $\hat{r}_i \leftarrow \hat{r}(traj_i)$

            $\hat{r}_j \leftarrow \hat{r}(traj_j)$

            $loss \leftarrow CrossEntropyLoss(\hat{r}_i, \hat{r}_j, label)$

            $loss.backward()$

        **end for**

        $loss_{val} \leftarrow$ calculate cross-entropy loss on validation set

        **if** $loss_{val}$ doesn't decrease for 10 epochs **then**

            **break**

        **end if**

    **end for**

    $\hat{\pi} \leftarrow$ Run RL (PPO or SAC) using $\hat{r}$ as reward for 1M iterations

**Output:** $\hat{r}, \hat{\pi}$

---

## A.2 SYNTHETIC PREFERENCE GENERATION

To enhance scalability and reproducibility, we automatically generate a large amount of synthetic trajectory preferences. This was done using an expert RL policy trained using the ground-truth reward function provided with each of environment. We then generate a large number of diverse trajectories by adding $\epsilon$-greedy noise during policy rollouts, where $\epsilon$ is the probability that the policy takes an action uniformly at random from its action space. Thus, $\epsilon = 0$ corresponds to the fully trained RL policy and $\epsilon = 1$ corresponds to a uniformly random policy. As noted by Brown et al. (2020b), adding this type of disturbance will result in monotonically decreasing performance in expectation.

To generate pairwise preferences over trajectories, we select all pairs of trajectories from a set of 40, 120, and 324 total trajectories (for the SMALL, MEDIUM, and LARGE dataset sizes, respectively) generated with $\epsilon$-greedy rollouts for $\epsilon \in \{0, 0.2, 0.4, 0.6, 0.8, 1\}$. We use held-out sets of trajectories for validation and testing. We then use the ground-truth reward functions provided by each environment to provide ground-truth preference labels.

## A.3 EVIDENCE OF CAUSAL CONFUSION

Table 2 displays results when dataset size is varied (S, M, L). Table 3 displays additional results for the Half Cheetah environment (Brockman et al., 2016), where we notice that, unlike the other environments presented, results vary depending on dataset size. We observe the same pattern *in the S data size* of high reward model performance on a held-out set, low subsequent policy (PREF) performance when the reward model is optimized, and the reward model's preference for the learned suboptimal policy rather than the expert policy (GT). However, *in the M and L dataset sizes*, we observe that the learned reward is more or less aligned with the true reward–it recognizes that the GT policy is better (L) or about the same (M). Thus, the poor performance of the policy in HalfCheetah (L) is more due to a failure in RL, and one could reasonably expect that running RL for more carefully-tuned hyperparameters and iterations on this reward model would result in good performance. This agrees with the HalfCheetah results in Brown et al. (2019). Nonetheless, we maintain that this is due to the simplicity of the true HalfCheetah reward (with which preferences are generated), which essentially only depends on one feature: the x-velocity (which the reward model has explicit access to as one of the features in the observation).

Fig. 8 provides plots of select features' spurious correlations with the true reward. Fig. 9 shows gradient saliency maps of the learned rewards that elucidate how the learned reward may improperly weight a feature.

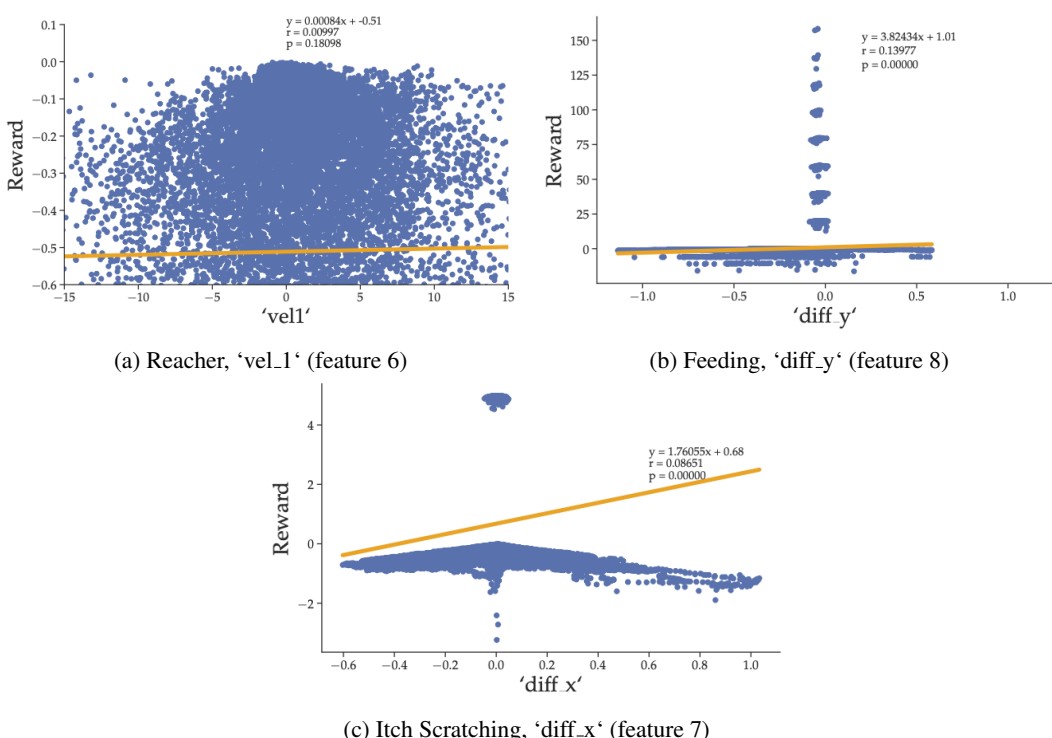

(a) Reacher, 'vel_1' (feature 6)

(b) Feeding, 'diff_y' (feature 8)

(c) Itch Scratching, 'diff_x' (feature 7)

Figure 8: **Correlations of Spuriously Weighted Features with Reward.**

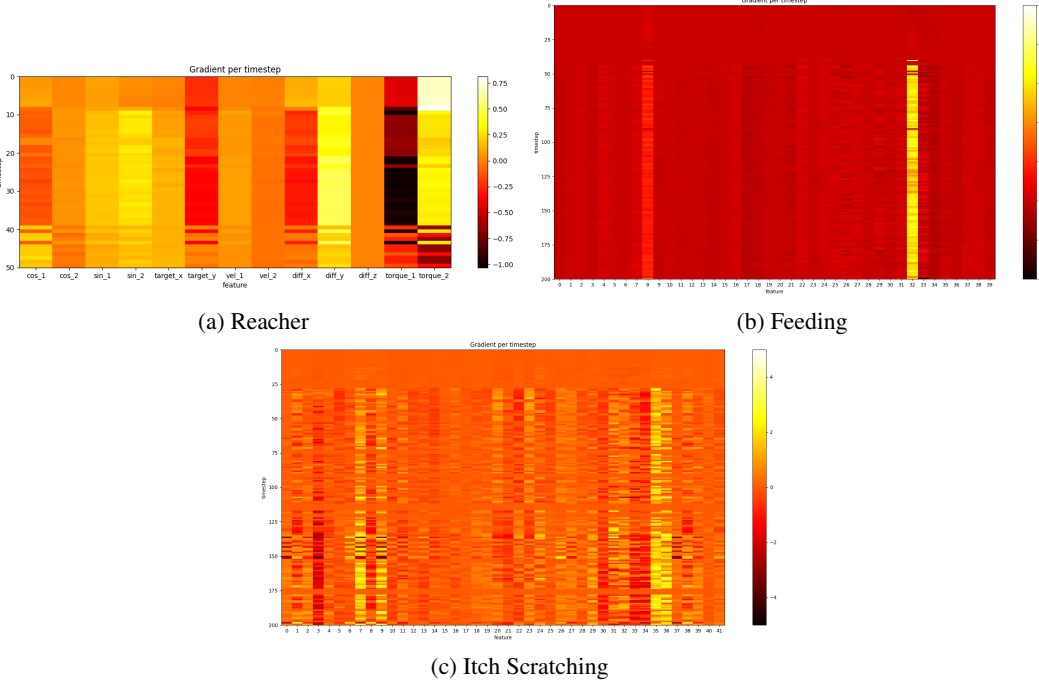

(a) Reacher

(b) Feeding

(c) Itch Scratching

Figure 9: **Gradient Saliency Maps for Fig. 2**. Fig. 9a shows the slight positive gradient for the angular velocity feature 'vel_1', which results in spinning behavior. Fig. 9b shows the nearly constant positive gradient in the 'diff_y' feature (feature 8) that incentivizes the robot to move into, past, and behind the human's head. A properly learned reward would exhibit a gradient that 'flips' to negative the moment the robot arm goes behind the human's head. Fig. 9c shows significant positive gradients for features 7, 9, 35, and 36, which correspond to 'diff$'_x$,' $diff'_z$, and the last two degrees of freedom in the $7 - DOF$ robot arm, respectively.

Table 2: **Empirical evidence of causal confusion (different dataset sizes).** We compare policies optimized with a reward learned from preferences (PREF) against policies optimized with the true reward (GT). State features on which preferences are based are fully-observable in all three tasks. S, M, and L correspond to training dataset sizes of 780, 7140, and 52326 unique pairwise preferences, respectively. Both PREF and GT are optimized with 1M RL iterations and averaged over 3 seeds. Despite high pairwise classification accuracy, the policy performance achieved by PREF under the true reward is very low compared with GT, irrespective of data size. However, *the reward learned from preferences consistently prefers PREF over GT.* This, combined with the low success rates of PREF compared to GT, suggests that preference-based reward learning fails to learn a good reward, even as the amount of data increases, for all of these tasks. (Lunar Lander does not have a predefined success metric, so we leave this column blank.)

| | PREF. LEARNING ACC. | | | RL POLICY PERFORMANCE | | |
|---|---|---|---|---|---|---|
| DOMAIN | TRAIN | VAL | TEST | LEARNED (PREF/GT) | TRUE (PREF/GT) | SUCCESS (PREF/GT) |
| REACHER (S) | 0.955 | 0.913 | 0.939 | -1.097 / -6.002 | -13.331 / -5.560 | 0.040 / 0.827 |
| REACHER (M) | 0.957 | 0.949 | 0.962 | -12.002 / -14.936 | -11.890 / -5.560 | 0.053 / 0.827 |
| REACHER (L) | 0.954 | 0.956 | **0.966** | **44.988** / 3.395 | -42.716 / **-5.560** | 0.100 / **0.827** |
| FEEDING (S) | 0.976 | 0.902 | 0.891 | 90.671 / 13.206 | -153.012 / 128.933 | 0.057 / 0.990 |
| FEEDING (M) | 0.979 | 0.968 | 0.960 | 106.415 / 68.835 | -45.427 / 128.933 | 0.437 / 0.990 |
| FEEDING (L) | 0.987 | 0.976 | **0.976** | **277.152** / 124.016 | -27.432 / **128.933** | 0.603 / **0.990** |
| ITCHING (S) | 0.974 | 0.908 | 0.869 | 18.757 / 10.337 | -56.591 / 248.397 | 0.000 / 0.970 |
| ITCHING (M) | 0.967 | 0.924 | 0.918 | 17.871 / 12.685 | -68.024 / 248.397 | 0.003 / 0.970 |
| ITCHING (L) | 0.954 | 0.933 | **0.928** | **16.588** / 10.282 | -47.190 / **248.397** | 0.013 / **0.970** |
| LUNAR (S) | 0.967 | 0.920 | 0.921 | 12.730 / 1.548 | -5496.430 / 178.617 | —- |
| LUNAR (M) | 0.947 | 0.945 | 0.937 | 73.342 / -4.511 | -6272.095 / 178.617 | —- |
| LUNAR (L) | 0.940 | 0.948 | **0.943** | **-1.375** / -6.307 | -747.081 / **178.617** | —- |

Table 3: **Empirical evidence of causal confusion (addt'l.).** We observe that the the learned reward (LEARNED) is more or less aligned with the true reward (TRUE) in the M and L dataset sizes and only really misidentified in S.

| | PREF. ACC. | | RL POLICY PERFORMANCE | | |
|---|---|---|---|---|---|
| DOMAIN | TRAIN | VAL | LEARNED (PREF/GT) | TRUE (PREF/GT) | SUCCESS (PREF/GT) |
| HALFCHEETAH (S) | 1.000 | 0.971 | 1608.492 / 1101.144 | 1626.549 / 3446.618 | 0.397 / 0.860 |
| HALFCHEETAH (M) | 0.999 | 0.993 | 684.118 / 676.197 | 2968.347 / 3446.618 | 0.647 / 0.860 |
| HALFCHEETAH (L) | 0.999 | 0.997 | 190.574 / 815.975 | 1236.655 / 3446.618 | 0.330 / 0.860 |

## A.4 INPUT FEATURES FOR ENVIRONMENTS

Features that are distractors are labeled with a *D*. Features that are causal are labeled with a *C*. The dimensions of each feature are included in parentheses. Lunar Lander:

- C - $x$-coordinates of the lander (1)
- C - $y$-coordinates of the lander (1)
- C - Linear velocity in $x$ (1)
- C - Linear velocity in $y$ (1)
- C - Angle of the lander (1)
- D - Angular velocity of the lander (1)
- C - Whether each leg is in contact with ground (2)
- C - Action — one of {do nothing, fire left orientation engine, fire main engine, fire right orientation engine} (1)

Reacher:

- D - Cos of angle of first and second arm (2)
- D - Sin of angle of first and second arm (2)
- D - Coordinates of target (2)
- D - Angular velocity of first and second arm (2)
- C - Position_fingertip - position_target (3)
- C - Action — Torque applied at first and second hinge (2)

Feeding:

- D - Spoon_pos_real (3)
- D - Spoon_orient_real (4)
- C - spoon_pos_real - target_pos_real (3)
- D - Robot_joint_angles (7)
- D - Head_pos_real (3)
- D - Head_orient_real (4)
- C - Spoon_force_on_human (1)
- C - Action (7)
- C - Foods_in_mouth (1)
- C - Foods_on_floor (1)
- C - Foods_hit_human (1)
- C - Sum_food_mouth_velocities (1)
- C - Prev_spoon_pos_real (3)
- C - Robot_force_on_human (1)

Itch Scratching:

- C - Tool_pos_real (3)
- D - Tool_orient_real (4)
- C - Tool_pos_real - target_pos_real (3)
- D - Target_pos_real (3)
- D - Robot_joint_angles (7)
- D - Shoulder_pos_real (3)
- D - Elbow_pos_real (3)
- D - Wrist_pos_real (3)
- C - Tool_force (1)
- C - Action (7)
- C - Prev_tool_pos_real (3)
- C - Robot_force_on_human (1)
- C - Prev_tool_force (1)

## A.5 GROUND TRUTH REWARD FUNCTIONS FOR ENVIRONMENTS

We outline the ground truth reward functions for each environment below. We also refer the reader to the publicly available code repositories for each environment, which describe the rewards in more detail.

Lunar Lander (`https://github.com/openai/gym`):

$$S(x) = -100\sqrt{(position_x^2 + position_y^2)}$$
$$+ -100\sqrt{(velocity_x^2 + velocity_y^2)}$$
$$+ -100|angle|$$
$$+ 10(contact_{left}) + 10(contact_{right})$$
$$R(x) = S(x) - S_{prev}(x)$$
$$R(x) = R(x) - 0.3(action)$$

Reacher (`https://github.com/openai/gym`):

$$R(x) = -||position_{fingertip} - position_{target}||_2 + -||action||_2^2$$

Feeding (`https://github.com/Healthcare-Robotics/assistive-gym`):

$$r_{distance} = -||pos_{target} - pos_{spoon}||_2$$
$$r_{action} = -||action||_2$$
$$r_{food} = f(Foods\_in\_mouth, Foods\_on\_floor)$$
$$preferences\_score = g(Foods\_hit\_human, Sum\_food\_mouth\_velocities,$$
$$Spoon\_pos\_real, Prev\_spoon\_pos\_real, Robot\_force\_on\_human)$$
$$R(x) = W_{distance} * r_{distance} + W_{action} * r_{action} + W_{food} * r_{food} + preferences\_score$$

Itch Scratching (`https://github.com/Healthcare-Robotics/assistive-gym`):

$$r_{distance} = -||pos_{target} - pos_{spoon}||_2$$
$$r_{action} = -||action||_2$$
$$r_{scratch} = f(Tool\_pos\_real, Target\_pos\_real, Prev\_tool\_pos\_real,$$
$$Tool\_force, Prev\_tool\_force)$$
$$preferences\_score = g(Spoon\_pos\_real, Prev\_spoon\_pos\_real,$$
$$Robot\_force\_on\_human, Tool\_force, Target\_pos\_real)$$
$$R(x) = W_{distance} * r_{distance} + W_{action} * r_{action} + W_{scratch} * r_{food} + preferences\_score$$

## A.6 EVALUATING LEARNED REWARD FUNCTIONS

***Saliency maps*** are one of the few methods that allow one to interpret learned reward functions in an isolated, relatively lightweight manner. Following Michaud et al. (2020), we use raw gradient saliencies, or $\frac{\partial R}{\partial(s,a)}$—the gradient with respect to each element of the input. We extend upon this by examining gradient saliencies *per timestep* along with feature *spread maps*—maps of each input feature's variation (standard deviation, variance, range) over the course of a trajectory.

We produce saliency maps of the learned reward as follows: we forward propagate a single rollout from the learned reward's policy through the reward network. Then, with the reward model's weights fixed, we backpropagate the output from the forward pass through the network and into the input (the policy rollout) to obtain the gradient with respect to each feature in the observation-action pair at each timestep.

***EPIC***, or Equivalent-Policy Invariant Comparison, is a pseudometric proposed by Gleave et al. (2020) that quantifies the difference between two reward functions on a given coverage distribution and proposes to be predictive of policy performance without the need for policy optimization. We apply the EPIC pseudometric to compare the distance between various learned rewards and the

Table 4: **KL Divergence: Distractor Features.**

| ENV | WITH | WITHOUT |
|---|---|---|
| FEEDING | 4.732 | **6.985** |
| ITCH SCRATCHING | **18.749** | 8.570 |

ground truth reward on the coverage distributions seen during reward learning and those seen during policy training. Using other metrics to compare learned rewards (such as DARD, by Wulfe et al. (2022)) may also prove fruitful in future work.

***Kullback-Leibler divergence***: We use a discriminator trained to minimize the cross-entropy loss on states from two different distributions following the approach proposed by Huszár (2017) and Laidlaw & Dragan (2021). Specifically, we approximate the KL divergence between the distribution of state-action pairs seen during reward learning and those seen during RL on the learned reward. We use this to measure the amount of distribution shift from the reward learning distribution induced by optimizing (potentially misidentified) learned rewards during policy training.

Specifically, for each learned reward, we sample 50 trajectories from the reward's training data and from the resulting policy (taking care to label each trajectory's origin distribution). From these 50x2 trajectories, we create the training and validation splits and then flatten each group of trajectories into a dataset of observation-action pairs. We train a discriminator model (hidden dimensions of 128x128x128) to distinguish between observation-action pairs seen during reward learning and those during RL by minimizing the binary cross-entropy loss. We tune hyperparameters (learning rate, weight decay) on the validation loss and accuracy.

With this trained discriminator model, we calculate $D_{KL}(p||q)$ by taking the discriminator's negative mean return/logit of all reward learning observation-action pairs, where $p$ is the reward learning distribution and $q$ is the policy optimization distribution. Similarly, we calculate $D_{KL}(q||p)$ by taking the mean return of all RL observation-action pairs. Since the KL divergence is not symmetric, we report $D_{KL}(p||q) + D_{KL}(q||p)$, or the symmetric KL divergence. For a proof on why a discriminator can be used to approximate the KL divergence, we refer the reader to Appendix B.2 of Laidlaw & Dragan (2021).

### A.7 DISTRACTOR FEATURES

Fig. 10 shows the additional results for removing distractors. Fig. 11 shows the saliency map for the Reacher learned reward without distractors. Table 4 shows additional KL divergences for the Feeding and Itch Scratching environments. We note here that the reason removing non-causal distractor features appears to not benefit for Feeding is that the main spurious correlation (discussed in Section 4) involves one of the *causal* features—namely, the difference between spoon position and target position. Thus, removing purely non-causal features fails to address this issue in Feeding.

### A.8 MODEL CAPACITY

We study the effect of model capacity on the learned reward. We find that, despite careful tuning of hyperparameters with each model and dataset size, increasing the capacity of the reward model does not necessarily result in an increase in subsequent policy performance. In fact, as seen in Fig. 12, only the Feeding task trained with the large (L) dataset size (52326 preferences) benefits from steadily increasing reward model capacity. Indeed, although increasing the capacity for the Reacher task appears to initially increase performance for the large dataset case (Fig. 12a), performance drops back down to below the performance of the smallest model size when the model size is further increased. Further, increasing model size seems to decrease the performance on the small datasets. This is not surprising, as benefiting from larger capacity tends to require increasing the amount of data. As such, the validation-set accuracies here tend to agree with learned reward performance.

To examine these results further, we compare the learned rewards directly with the ground truth reward (without performing policy optimization) using the EPIC pseudometric in Table 5. Inter-

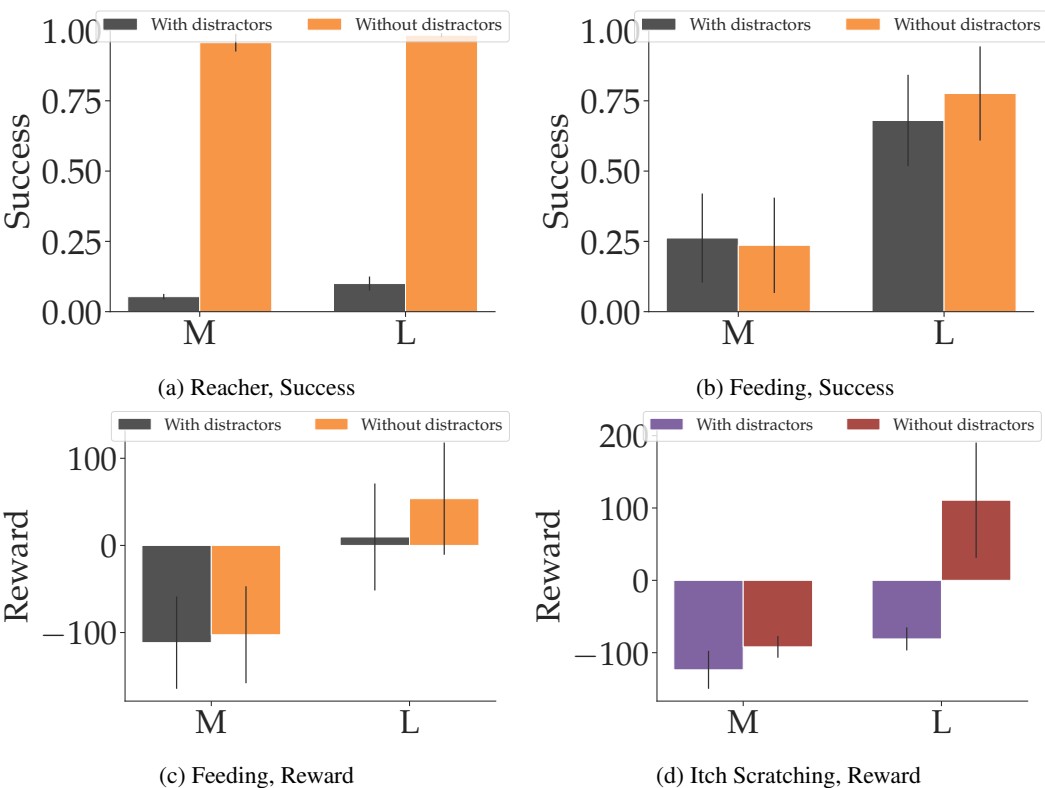

Figure 10: **Distractor Features.** In general, removing distractor features that have no influence on preferences improves performance.

Table 5: **EPIC Distances, Model Capacity.** The reward model with larger capacity (hidden layer dimensions of 256x256x256) appears closer to the ground truth than the reward model with smaller capacity (hidden layer dimensions of 128x64) *on the distribution of states seen during reward learning*, but is much further from the ground truth *on the distribution of states seen during reinforcement learning*.

| REWL / RL | 256x256x256 | 128x64 |
|---|---|---|
| GROUND TRUTH | 0.210 / 0.707 | 0.233 / 0.598 |

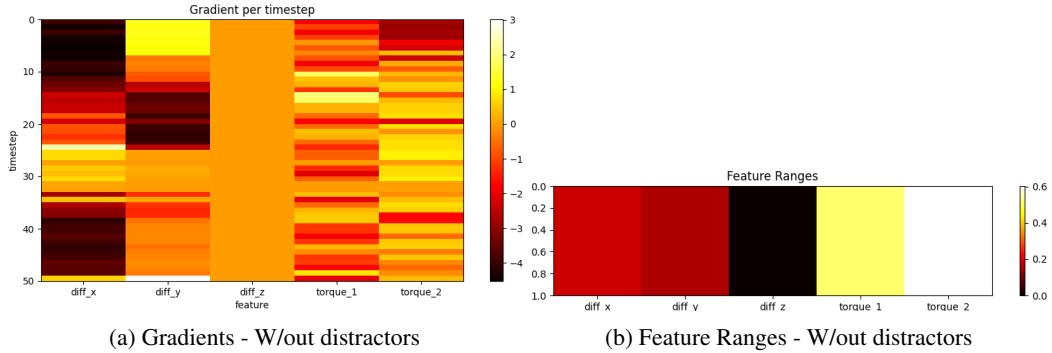

(a) Gradients - W/out distractors  (b) Feature Ranges - W/out distractors

Figure 11: **Saliency Maps: Without Distractor Features.**

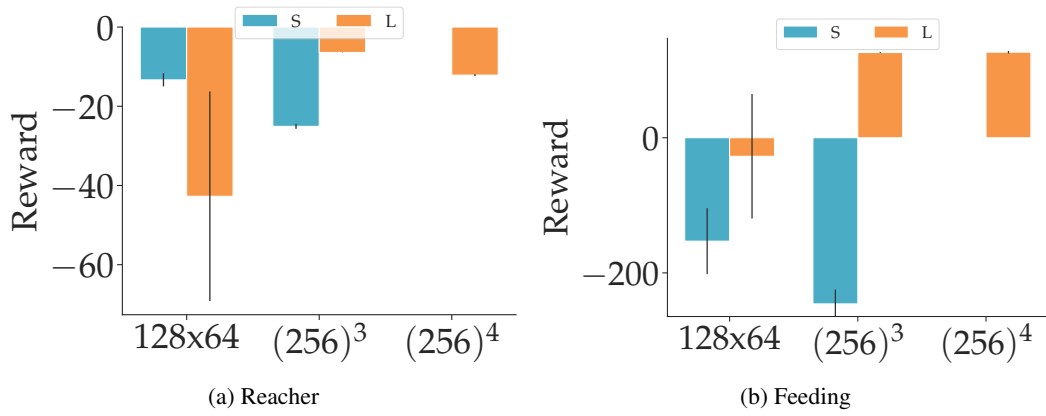

(a) Reacher  (b) Feeding

Figure 12: **Model Capacity.** Increasing model capacity increases performance when we use the large (L) dataset ($\binom{324}{2} = 52326$ preferences), to a certain extent. However, increasing capacity decreases performance on the small (S) dataset ($\binom{40}{2} = 780$ preferences).

estingly, the EPIC distances between the learned rewards and the ground truth reward vary widely depending on the choice of coverage distribution. With the distribution of states seen during reward learning (generated by randomly switching between an expert and random policy), the larger (256x256x256) model appears closer to the ground truth reward than the smaller (128x64) one. However, these results are flipped when we instead evaluate the pseudometric on the distribution of states seen during policy optimization. This suggests that increasing the capacity of the reward model allows it to more closely 'mimic' the ground truth reward *on the reward learning distribution* (hence the lower EPIC distance) but worsens its ability to generalize to the reinforcement learning distribution. We see a similar effect with the divergence metric in Table 6. .

## A.9 NOISE IN STATED PREFERENCES

Fig. 13 shows additional results. Increasing the amount of preference data when noise is present appears to have a negative effect (as seen in Fig. 13b), despite the proportion of mislabeled data

Table 6: **KL Divergence Approximations, Model Capacity.** Increasing model capacity may result in a much larger distribution shift (as measured by the KL Divergence).

|  | 128x64 | 256x256x256 | 128x64 | 256x256x256 | 256x256x256x256 |
|---|---|---|---|---|---|
| REACHER | 64.165 | 57.724 | 46.031 | 22.187 | 88.927 |
| FEEDING | 25.352 | 32.836 | 16.900 | 4.732 | 9.909 |

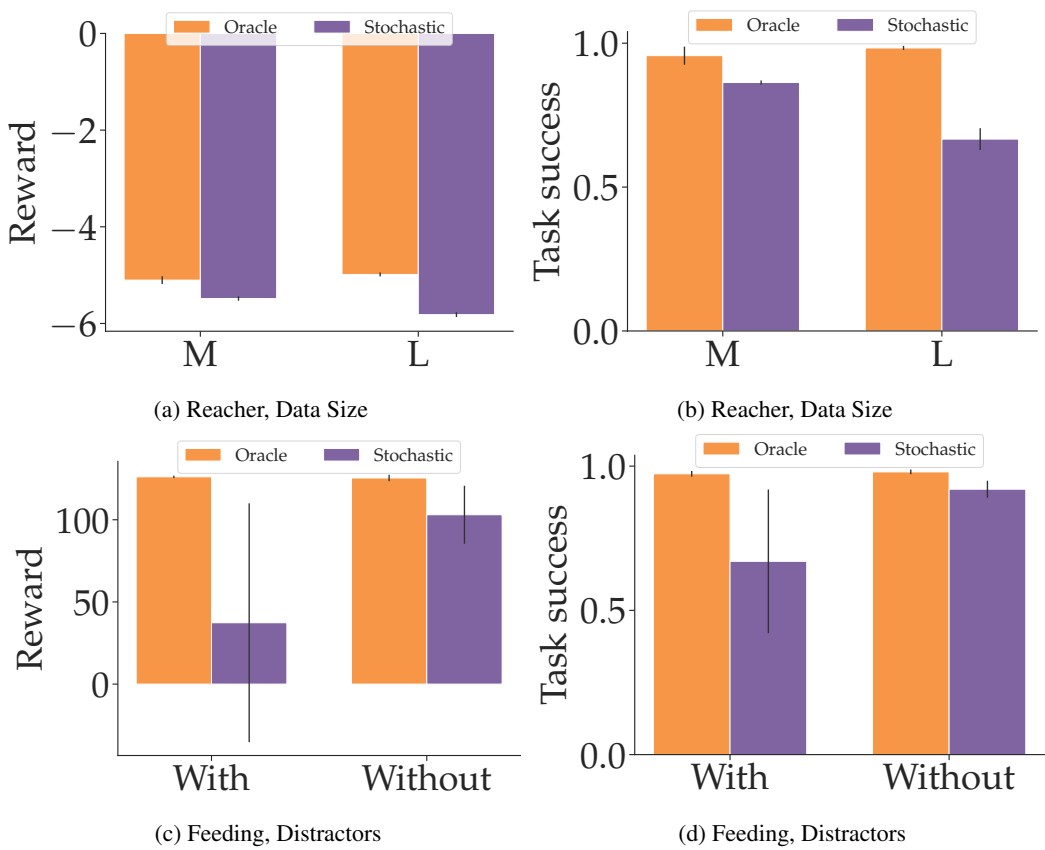

Figure 13: **Noise in Stated Preferences.**

Table 7: **Percentage (%) of Mislabeled Data Provided by a Stochastic User.** Numbers here correspond to Fig. 13b.

|  | MEDIUM | LARGE |
| --- | --- | --- |
| STOCHASTIC | 2.726 | 2.870 |

remaining constant across dataset sizes (Table 7). Further, there is a compounding effect with distractor features, where both noise and distractors result in a large loss in performance (Fig. 13d).

To examine these results further, we compare the learned rewards directly with the ground truth reward (without performing policy optimization) with the EPIC pseudometric in Table 9. Interestingly, the EPIC distances between the learned rewards and the ground truth reward vary depending on the choice of coverage distribution. With the distribution of observation-action pairs seen during reward learning (generated by randomly switching between an expert and random policy), the reward trained with labeling errors appears to be closer (in EPIC distance) to the ground truth than the reward trained without labeling errors. However, these results are flipped when we instead evaluate the pseudometric on the distribution of observation-action pairs seen during policy optimization. This suggests that the misidentified reward model more closely 'mimics' the ground truth reward *on the reward learning distribution* (hence the lower EPIC distance) but fails to generalize to the reinforcement learning distribution.

Table 8 shows the KL divergences for each model configuration in Fig. 5.

Table 8: **KL Divergence Approximations, Noise in Stated Preferences.**

|  | ORACLE | STOCHASTIC |
|---|---|---|
| REACHER, MEDIUM DATASET | 6.762 | **8.856** |
| REACHER, LARGE DATASET | 7.365 | 6.739 |
| FEEDING, WITH DISTRACTORS | 4.732 | **13.188** |
| FEEDING, WITHOUT DISTRACTORS | 6.985 | **8.046** |

Table 9: **EPIC Distances, Noise in Stated Preferences.** Misidentified rewards (Stochastic) are closer to the ground truth reward on the reward learning distribution (RewL) but further from it on the RL distribution (RL) when compared to the properly learned rewards (Oracle). EPIC distances are computed on the rewards learned with distractors in the Feeding environment in Fig. 13d and averaged over three seeds.

| REWL / RL | GROUND TRUTH | STOCHASTIC | ORACLE |
|---|---|---|---|
| GROUND TRUTH | 0.000 / 0.000 | **0.136 / 0.164** | **0.165 / 0.130** |
| STOCHASTIC | **0.136 / 0.164** | 0.000 / 0.000 | 0.220 / 0.209 |
| ORACLE | **0.165 / 0.130** | 0.220 / 0.209 | 0.000 / 0.000 |

## A.10 PARTIAL OBSERVABILITY OF CAUSAL FEATURES

Fig. 14 displays gradient maps of a reward that partially observes the causal features and a reward that fully observes the causal features. Table 10 displays the EPIC distances. Unlike with previous factors, EPIC distance on the reward learning distribution is predictive of eventual policy performance. The KL divergences (Table 11) again reveal that the misidentified reward (this time, due to lack of information in the observation) leads to a greater distribution shift during RL training.

We see that without having access to all the causal reward features, an incorrect reward model is learned due to over-weighting the causal features that are available and also due to spurious correlations with non-causal variables. With Feeding, we observe both cases: the reward model learns stronger weights on one of the joint angles and two components of the action ('act'), all of which have no bearing on the true reward. Simultaneously, the reward also learns a greater weight on the second component of the feature corresponding to the vector distance between the end effector and the mouth ('diff'), which is causal in the ground truth reward. (Note that a similar weighting also occurs in the learned reward that has access to all features of the true reward.) This results in the behavior depicted in Fig. 6—the robot manipulator is able to successfully maneuver the spoon close to the patient's mouth (by observing the distance feature), but does so without ensuring that the food particles themselves stay on the spoon and end up in the patient's mouth (since the reward is unable to observe the number of food particles).

## A.11 COMPLEX CAUSAL FEATURES

In the Itch Scratching task rewards the robot needs to perform a 'scratching' motion, which entails not only making contact with the target using the end-effector (being within a certain radius of the target coordinates), but also requires that the target contact position be greater than a certain $\delta$ away from the previous target contact position and that the exerted force be no more than a specified

Table 10: **EPIC Distances, Partial Observability.**

| REWL / RL | GROUND TRUTH | PARTIAL | FULL |
|---|---|---|---|
| GROUND TRUTH | 0.000 / 0.000 | 0.684 / 0.699 | 0.165 / 0.309 |
| PARTIAL | 0.684 / 0.699 | 0.000 / 0.000 | 0.685 / 0.697 |
| FULL | 0.165 / 0.309 | 0.685 / 0.697 | 0.000 / 0.000 |

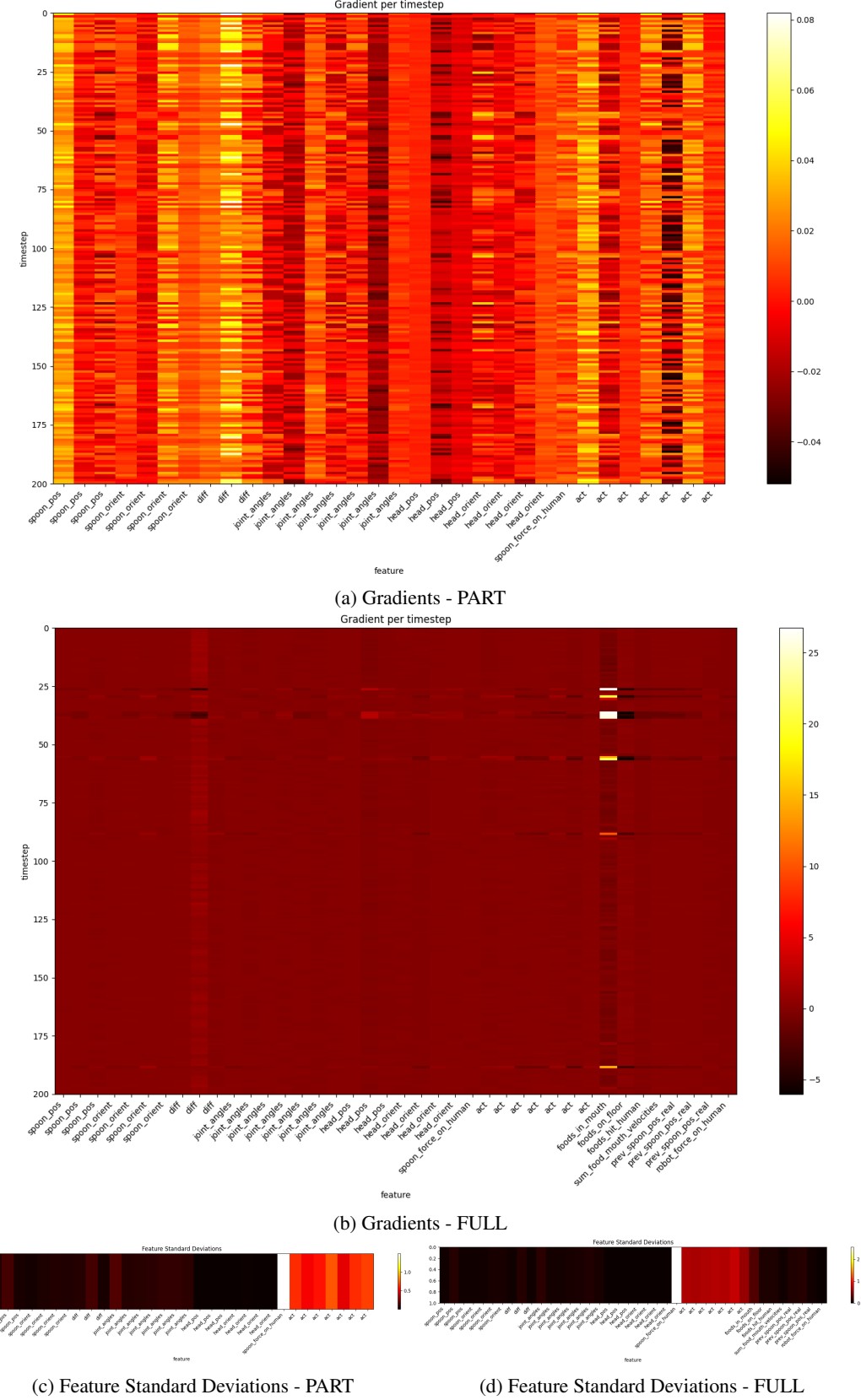

Figure 14: **Saliency Maps, Partial Observability.**

Table 11: **KL divergence: partial observability.** The reward with partial observability on causal features induces a greater distribution shift than the reward with full observability.

| | FULL | PARTIAL |
|---|---|---|
| FEEDING | 4.732 | **45.731** |

Table 12: **KL Divergence: Complex Causal Features.** Learning a reward without the 'scratched' indicator feature leads to a greater amount of distribution shift (KL divergence) during policy optimization.

| WITH 'SCRATCHED' | WITHOUT 'SCRATCHED' |
|---|---|
| 8.570 | **15.553** |

$F_{max}$. Although the preferences are based on how well the robot performs this scratching motion and despite the reward model having access to all the necessary low-level information to infer this scratching motion (including information about the state at the previous timestep; see Appx. A.4 for details), we find that the reward model is not able to learn the scratching motion. As seen in Fig. 7c, once we explicitly include a high-level indicator feature, 'scratched', denoting whether the robot has successfully performed the aforementioned scratching motion, performance drastically increases. We suspect that the reward model's tendency to pick up on spurious correlations that occur consistently over the course of the trajectory involving just a few variables prevents it from learning the true causal relation that involves many variables, each of which are causal only in a particular context. This is supported by Fig. 9c, which shows gradients staying relatively constant for each feature rather than varying with time and context. As a result, the learned reward without an explicit 'scratched' feature leads to a greater amount of distribution shift. This is shown in Table 12 which shows the KL divergences with and without the hand-specified complex 'scratch' feature.

### A.12 DATA GENERATION METHODS

Fig. 15 displays a comparison of our method of generating a diverse dataset of trajectory preferences and the method of using a checkpointed RL policy (proposed by (Brown et al., 2019)).

### A.13 PENALIZING THE KL DIVERGENCE

Inspired by the results showing that misidentified rewards result in greater distribution shifts during policy optimization, we attempt to address this by adding a KL divergence penalty term to the reward. Specifically, in addition to using the reward learned from preferences, we incorporate a pretrained discriminator (trained to discriminate between state-action pairs from the reward learning distribution and the RL distribution) to estimate the KL divergence, as detailed earlier in Section 3 and Appx. A.6. This KL divergence is scaled by a hyperparameter $\lambda$ and subtracted from the reward learned from preferences.

We show results for $\lambda = 0, 0.01, 0.1, 1, 10$ in Fig. 16. Using a discriminator that is pretrained offline (the discriminator isn't updated during the RL process) to penalize the KL divergence does not appear to help the performance or alleviate reward misidentification in any way. It appears that the RL agent is again able to hack the reward—this time, the performance and reward misidentification are even worse because of the additional degree of freedom in the reward function afforded by the discriminator penalty.

### A.14 ITERATIVE PREFERENCE-BASED REWARD LEARNING FROM ONLINE DATA

Following Christiano et al. (2017), we verify the occurrence of reward misidentification when learning from data that is actively queried online (during the learning process). As before, we use the initial set of offline trajectory and preference data (acquired via Appx. A.2) to train a reward function, which we then use to optimize a policy using RL. We then sample 10 trajectories from the most

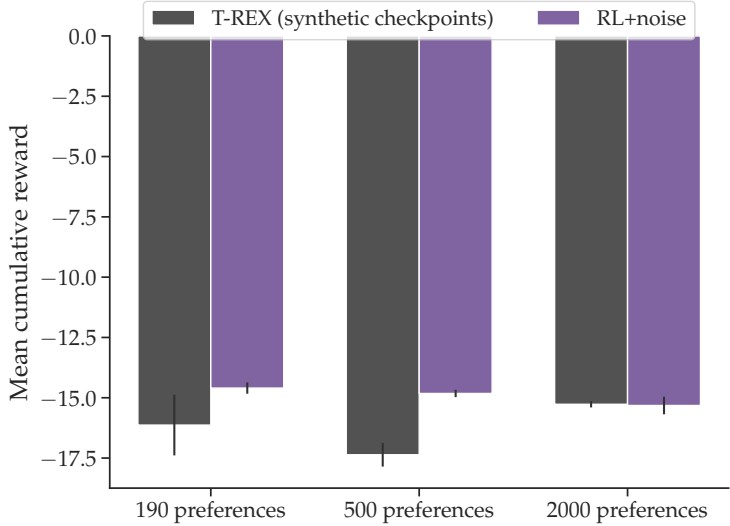

Figure 15: **TREX vs. RL+Noise.** Our method of generating diverse trajectories for preference learning performs on par with, if not better than, the method of using rollouts taken from a check-pointed RL policy, as proposed by (Brown et al., 2019). Displayed are cumulative trajectory rewards from the Reacher environment.

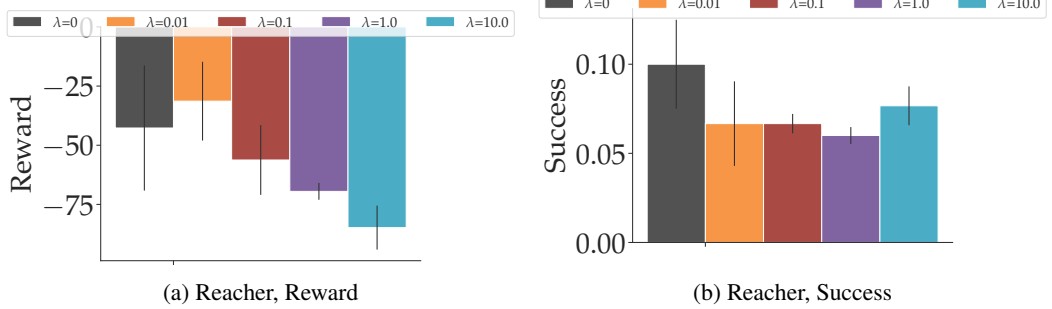

(a) Reacher, Reward

(b) Reacher, Success

Figure 16: **Penalizing the KL Divergence.**

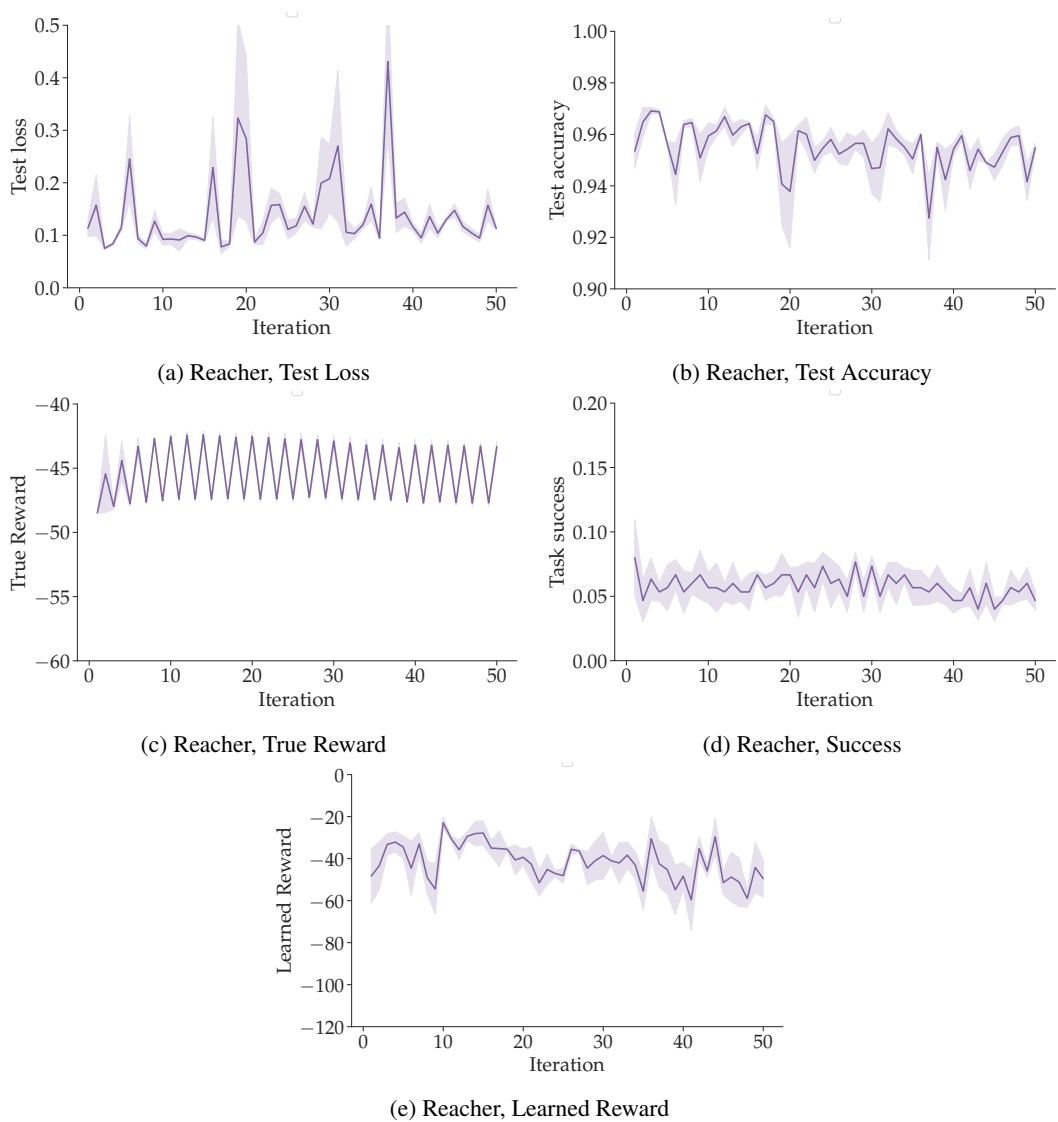

(a) Reacher, Test Loss

(b) Reacher, Test Accuracy

(c) Reacher, True Reward

(d) Reacher, Success

(e) Reacher, Learned Reward

Figure 17: **Training the reward and policy together iteratively.**

recently optimized policy and generate preferences for all possible pairs (1) between sampled and offline trajectories and (2) within the set of sampled trajectories, which we concatenate to our existing training dataset. Using the (now augmented) dataset, we fine-tune the reward function and policy for 10 epochs and 100 iterations, respectively. We repeat this process of taking rollouts from the most recent policy and fine-tuning the existing learned reward and policy for 50 iterations. Results are displayed in Fig. 17. We observe that reward misidentification appears to still be present—test error is very low, while resulting policy performance (measured by true reward and success) remains poor.

