# OpenReview forum: "Causal Confusion and Reward Misidentification in Preference-Based Reward Learning"
_ICLR.cc/2023/Conference — ICLR 2023 poster_

### Official Review · Reviewer_VP1L · 2022-10-21

**Confidence:** 3
**Correctness:** 4
**Technical Novelty And Significance:** 2
**Empirical Novelty And Significance:** 3
**Recommendation:** 8

**Clarity, Quality, Novelty And Reproducibility:**

When creating the synthetic preference dataset, varying amounts of noise is added to the behaviors from the policy rollout.  When optimizing the reward do you sample behaviors across all noise levels?   If so I would be interested in seeing how this compares to iteratively optimizing the policy and the reward as over time the quality of the policy and reward improves, and so there should be less noise further in training.  How would this impact training and the trained models?

It would be useful to include an overview of the algorithm in an appendix to help properly understand the interaction between the policies and the rewards during optimization — e.g., confirm that you do not alternate between policy updates and reward updates, and to specify how the behaviors are sampled — e.g., randomly drawn from all noisy levels at each step, or begin with \epsilon=1 and decay towards \epsilon=0 to simulate the performance of the policy training?

Nit-pick:  throughout the paper, the incorrect opening quote is used.


**Strength And Weaknesses:**

+The paper drives home the message that being able to predict preferences does not necessarily translate to quality task completion.
+The experiments are thorough and consider various issues that could be the cause of the failure with learned reward functions.
+The test environments offer a range of difficulty, and the findings are consistent across tasks.

-The paper reads as though the policies and rewards are trained independently.  First train a policy using the ground-truth environment reward, sample behaviors and add noise to form the dataset for reward learning, optimize the learned reward function, then finally train a new policy on this reward.  However, as noted in Cristiano et al. (Section 2.2), it is important the the policy and the reward are learned together.
-The number of pairs in the datasets is somewhat large.  For Table 1, the large dataset is >50k pairs.  For a real problem this is too large to imagine being labelled by people.

**Summary Of The Paper:**

This paper provides an analysis of reinforcement learning reward functions learned from preference labels over trajectory pairs.  In particular, the authors show that such learned reward functions are prone to pick up on spurious correlations in the features, to noise in the preference labels, and to distribution shifts when optimizing the reward model.  Furthermore, the authors demonstrate that is not simply an over-fitting problem as reward models can reliably predict the preferred trajectory within a pair of trajectories in validation and test data, but rather that the policy is unable to exploit the reward as the policy shifts from the training distribution.

**Summary Of The Review:**

Preference-based reward learning is becoming increasing popular, especially for domains in which it is challenging to hand-define a reward function.  Understanding the impact and limitations of learned rewards is important, and the work in this paper provides insights towards that.

I have some concern about the flow of the training — it’s not that what is done here is wrong, but rather there are better ways to train the policies and rewards together.  However, with the authors open sourcing their code, others can extend the work to consider how they specifically go about training.

---

> ### Author Response · Authors · 2022-11-16
> **Response to Reviewer VP1L**
>
> Thank you so much for the helpful feedback! We've addressed your concerns in the revised paper (updates in blue) and respond to them below.
>
> > As noted in Cristiano et al. (Section 2.2), it is important the the policy and the reward are learned together.
>
> We agree that active learning is one potential way to address this problem setting and one that we mention in the conclusion. However, we focus on the recently popularized offline imitation learning setting [3,4,5], where, as done in prior work, we learn the reward function from offline preferences [1,2] and analyze the interesting causal misidentification problems that can occur from learning from only offline pairwise preferences. Nonetheless, we include an exploration of Christiano et al.’s method of iteratively learning the reward and policy in Appendix 13.
>
> [1] Brown, Daniel, et al. "Extrapolating beyond suboptimal demonstrations via inverse reinforcement learning from observations." International conference on machine learning. PMLR, 2019.
>
> [2] Wang, Ruohan, et al. "Random expert distillation: Imitation learning via expert policy support estimation." International Conference on Machine Learning. PMLR, 2019.
>
> [3] Kim, Geon-Hyeong, et al. "DemoDICE: Offline imitation learning with supplementary imperfect demonstrations." International Conference on Learning Representations. 2021.
>
> [4] Jiang, Shengyi, Jingcheng Pang, and Yang Yu. "Offline imitation learning with a misspecified simulator." Advances in neural information processing systems 33 (2020): 8510-8520.
>
> [5] Xu, Haoran, et al. "Discriminator-weighted offline imitation learning from suboptimal demonstrations." International Conference on Machine Learning. PMLR, 2022.
>
>
>
> > The number of pairs in the datasets is somewhat large. For Table 1, the large dataset is >50k pairs. For a real problem this is too large to imagine being labelled by people.
>
> We agree that the size of the large dataset is exaggerated—our point was to have so many pairwise preferences that things "should" work. The fact that it doesn't provides evidence of a problem. We do observe similar/somewhat more drastic causal confusion with fewer preference pairs (Table 1; success decreases with less data, for example).
>
> > When optimizing the reward do you sample behaviors across all noise levels? If so I would be interested in seeing how this compares to iteratively optimizing the policy and the reward as over time the quality of the policy and reward improves, and so there should be less noise further in training. How would this impact training and the trained models?
>
> Yes, we sample an equal number of trajectories for each noise level. It's a good point that iteratively optimizing the policy and the reward over time should decrease the equivalent “noise level” of the data. However, as described in Appendix A13, it appears that, even so, the performance is roughly the same as without the iterative updates (albeit with decreased variance).
>
> > It would be useful to include an overview of the algorithm in an appendix to help properly understand the interaction between the policies and the rewards during optimization — e.g., confirm that you do not alternate between policy updates and reward updates, and to specify how the behaviors are sampled — e.g., randomly drawn from all noisy levels at each step, or begin with \epsilon=1 and decay towards \epsilon=0 to simulate the performance of the policy training?
>
> Thank you for the suggestion. We added an overview of the algorithm in Appendix A1 and clarified that the preference data is generated offline. To clarify, for the main body of the paper, we do not alternate between policy updates and reward updates, and we sample trajectories from all noise levels at the very beginning—before reward learning or policy optimization.

---

> > ### Comment · Reviewer_VP1L · 2022-11-17
> > **Thanks for clarifying**
> >
> > Thanks for clarifying, answering the questions, and the helpful additions to the paper.

---

### Official Review · Reviewer_Ek1w · 2022-10-25

**Confidence:** 3
**Correctness:** 3
**Technical Novelty And Significance:** 2
**Empirical Novelty And Significance:** 2
**Recommendation:** 5

**Clarity, Quality, Novelty And Reproducibility:**

## Clarity
Can be improved. First, there is no formal definition of causal confusion in preference-based reward learning. Second, some part of the logic is jumped and need more connections between sections. For example, this work provides evidence of causal confusion in section 3. and factors in section 4. But there is no clear connections between the evidence and the factors. Finally, there is no showcase on how to leverage the findings in the study to avoid causal confusion in reward learning, which redues the contribution of this work.


## Quality
From the study perspective, the quality is good. This study was conducted on three robot control benchmarks for preference-based reward learning, using diverse evaluating approaches and indentify several factors that lead to causal confusion.

## Novelty
The study is novel.

## Reproducibility
Need more details to reproduce the results.


**Strength And Weaknesses:**

## Strength

+ This paper provides the first concrete and systematic study of causal confusion for preference-based reward learning on three robot control benchmarks.

+ Several techniques including saliency maps, EPIC reward distance and Kullback-Leibler divergence were used for evaluating the learned reward functions. This study tried to provide evidence of causal confusion in preference-based reward learning, and identify several factors that lead to causaul confusion.


## Weakness

+ There is no formal definition of causal confusion in preference-based reward learning. How to distinguish between the cuasal confusion and other factors that result in similar performance in the benchmarks, for example, overfitting or distribution shift?

+ EPIC distance might no be a proper method for evaluating the learned reward in distribution shift cases. A recent work [1] has identify that EPIC avoids policy optimization, but in doing so requires computing reward values at transitions that may be impossible under the system dynamics. This is problematic for learned reward functions because it entails evaluating them outside of their training distribution, resulting in inaccurate reward values that we show can render EPIC ineffective at comparing rewards.

+ It is not clear how to use the findings in this study to avoid causal confusion in reward learning. It would be more interesting if this work can provides some suggestions or algorithms to do so.

[1] Wulfe, Blake, et al. "Dynamics-Aware Comparison of Learned Reward Functions." In ICLR 2022.







**Summary Of The Paper:**

This paper focus on causal confusion in preference-based reward learning. The study was conducted by performing a series of sensitivity and ablation studies on three robot control benchmarks, where the preference-based reward learning methods fail to generalize to out-of-distribution states.  The study indicated some sources that exacerbate causal confusion. Moreover, a set of methods were identified to interpret causally confused rewards.

**Summary Of The Review:**

Overall, this work provides a novel study for causal confusion in preference-based reward learning. To make it stronger, I suggest the authors to provide formal definition of causal confusion in reward learning, and showcase on how to leverage the findings in this study to avoid the causal confusion (if it is true) in reward learning.

---

> ### Author Response · Authors · 2022-11-16
> **Response to Reviewer Ek1w**
>
> Thank you for the insightful suggestions! We added a formal definition of causal confusion in preference-based reward learning and discuss how to leverage our findings to avoid causal confusion in our paper (edits in blue text). Detailed responses are below:
>
> > There is no formal definition of causal confusion in preference-based reward learning. How to distinguish between the causal confusion and other factors that result in similar performance in the benchmarks, for example, overfitting or distribution shift?
>
> Good point. Similar to “Causal Confusion in Imitation Learning” [1], we provide a definition of the ‘causal reward misidentification’ that we study in this paper in Section 2. We also added a causal diagram demonstrating the possible confounding that can happen and showing what is observed and unobserved.
>
> [1] De Haan, Pim, Dinesh Jayaraman, and Sergey Levine. "Causal confusion in imitation learning." Advances in Neural Information Processing Systems 32 (2019).
>
>
> > EPIC distance might no be a proper method for evaluating the learned reward in distribution shift cases. …
>
> We also found this to be somewhat problematic in our work—thanks for the pointer to DARD! (We added a reference to this in the future work section of our conclusion.) However, we ensure that all the transitions we sample are possible under the system dynamics by collecting transitions from a “mix” policy that randomly switches between taking actions from $\pi_{random}$ and $\pi_{expert}$. We also find that intentionally using EPIC to evaluate the same learned reward on different distributions produces enlightening results—the rewards that are causally confused often appear closer (in EPIC distance) to the true reward on the training distribution, but further from the true reward on the distribution induced by RL exploration, when compared with the properly learned reward.
>
> > It is not clear how to use the findings in this study to avoid causal confusion in reward learning. It would be more interesting if this work can provides some suggestions or algorithms to do so.
>
> Thank you for the suggestion. We agree that a discussion of potential solutions would strengthen the paper and have added this discussion as a new paragraph in the conclusion. We highlight ideas involving penalizing divergence from the training distribution, using human input to remove non-causal features, and using human input to better learn high-level causal features such as the “itch” feature described in our paper. We propose and explore two potential solutions: 1) penalizing the KL divergence during RL and 2) performing active learning by iteratively querying for online preferences. These are located in Appendices A12 and A13, respectively.
>
> > Some part of the logic is jumped and need more connections between sections. For example, this work provides evidence of causal confusion in section 3. and factors in section 4. But there is no clear connections between the evidence and the factors.
>
> Thanks for the feedback! We added some wording to make the transition between the ‘Evidence’ section and the ‘Factors’ section flow smoother at the beginning of Section 5: The factor of Distractor Features draws directly from the findings in the Evidence of Causal Confusion section, where we find that certain non-causal features may be spuriously correlated with the reward. The factor of Noise in Stated Preferences is inspired by the fact that preference data collected from real humans is often rife with different types of bias and noise. Partial Observability of Causal Features is also due to the fact that it may not always be possible to fully-observe all causal features in the real world. Lastly, Complex Causal Features is a topic that may become increasingly relevant in higher-dimensional, complex robotic tasks—unlike toy environments, where certain non-causal features can be ignored entirely, features here may be causal in certain contexts and non-causal in others.

---

> ### Author Response · Authors · 2022-11-17
> **Re: Response to Reviewer Ek1w**
>
> Dear reviewer Ek1w,
>
> We are grateful for your time and apologize for the intrusion, but as tomorrow is the last day we can update the draft, we are wondering if our response makes sense and if there are any other changes you’d like to see. Thank you!

---

### Official Review · Reviewer_hap7 · 2022-10-26

**Confidence:** 4
**Correctness:** 2
**Technical Novelty And Significance:** 3
**Empirical Novelty And Significance:** 3
**Recommendation:** 6

**Clarity, Quality, Novelty And Reproducibility:**

__Novelty__

This appears to be the first paper that focuses primarily on hacking of reward functions that are learned, not programmed. Some papers have studied hacking of learned reward functions as a side-note, and anecdotally identified causal confusion as the culprit but not studied it in detail. The authors should discuss the relationship to goal misgeneralization (Langosco et al and Shah et al) as these papers also empirically study goal/reward functions that lead to good performance on the training distribution but not elsewhere, due to spurious reward correlations (one difference is that their reward function is learned implicitly by the policy rather than with a reward model). The authors should also discuss to what extent the causal confusion problem in imitation learning is similar or different from the problem in preference learning. This would help me understand how novel the findings are.

__Quality__

Most claims appear to lack strong evidence.

Claim 1: reward hacking is due to spurious features that are being rewarded. This claim is the focus of Section 3.
- The first piece of evidence is that noise worsens final performance, and the paper suggests that this means causal confusion is the culprit. However, no explanation is given why independent noise should worsen causal confusion. (Previous work only gave an explanation for temporally correlated noise, but here the noise is independent.)
- The second piece of evidence is that removing non-causal features improves performance. This is valid but only shown in one environment (Reacher). In the other environments, results are unclear or not visible (Fig 7).
- The third piece of evidence is that a saliency map shows that features are rewarded that are not causally related to ground truth reward. However, it must also be shown that these features are correlated with reward on the training data, otherwise no spurious correlation has been shown. There may be a spurious correlation in the feeding environment since the spoon is never behind the head during training. However, for other environments it is not shown that the rewarded features actually correlate with reward on the training data.

Claim 2: a) preference noise worsens causal confusion and b) this becomes worse with more data. a) is not shown; the paper only seems to show that adding noise worsens task success, not causal confusion. b) is not given a convincing explanation, which means we need strong empirical evidence, ideally from more than two environments (Reacher and Feeding).

Claim 3: greater model size worsens causal confusion. Figure 9 shows that this sometimes happens and some times the opposite happens. I don't think there is enough evidence here to make a claim that belongs in the abstract.

Claim 4: when causal features are unobserved, causal confusion can get worse. This is shown in one environment (Feeding), which is acceptable since this claim is unsurprising.

To improve the paper, the authors could collect more evidence for each claim or reduce the number of claims and focus on the ones that matter most. I believe the paper would be strong enough by making only a subset of these claims, but supporting them well.

__Details__

Section 4.5: The experiment simply shows that the learned policy fails. However, it does not show that this is due to causal confusion, reward hacking, and/or distribution shift. So the value of this experiment is unclear.

Section 4.3: "increasing the data size without increasing data diversity results in data points that are more similar to each other" --> Why would trajectories become more similar when more samples are drawn but the distribution they are drawn from does not change? This claim needs to be supported or removed.

- The figure presentation could be easily improved.
    - 1) Figure 3 is not self-explanatory. Figures (incl. caption) should be self-explanatory if possible because most readers will skim them without reading the text, or skim the figures before reading the text.
    - 2) In Figure 1a, it is hard to understand what is happening in the top right image, especially what the meaning of the arrow is. It is also unclear if this is meant to show an OOD situation. Furthermore it should be clarified that the < refers to what preferences learned by the model, not ground truth. It should also be clarified that the right column refers to OOD situations which are caused by imperfections in the reward model, as this is not clear from the figure alone.
    - 3) Figure 1b is used in the introduction but only appears on page 5; ideally it would appear much earlier.
    - 4) Figure 1c: again, the meaning of the arrow is unclear without reading the main text. Perhaps use a circular arrow to show rotation.
    - 5) Table 2: the first sentence of caption is ambiguous and therefore difficult to follow, please rewrite. Furthermore, the text references Table 2c which does not exist.
    - 6) In Figure 5 it is hard to see the food, perhaps zoom in.

Lastly, I was unsure what the main contributions are. Section 3 suggests that a key contribution is to show that the reason for reward hacking is causal confusion. This is also implied in the introduction: "we demonstrate the failure of preference-based reward learning to produce causal rewards". However, the abstract does not mention this contribution.

The presentation is adequate but the paper tries to prove too many things which can be confusing.


**Strength And Weaknesses:**

RL from preferences has increasingly high-impact applications and great promise to be scaled further. This paper addresses reward hacking which is a common and severe difficulty with RL from feedback. Therefore it is important to understand the causes of this reward hacking. This paper identifies and analyzes one cause: spurious features that correlate with reward during training, are therefore rewarded by the reward model, which leads to troublesome behavior when training a policy with this reward. The paper also identifies exacerbating factors. Showing empirical evidence for these is a useful contribution since the problem is important and understudied.

However, the evidence for key findings is limited, which makes it hard to know how much the findings will generalize to other settings. Some claims also lack an explanation. I expand on this below.

**Summary Of The Paper:**

The paper studies reward hacking that happens when a policy is trained using a reward model that is learned from human preference data. In reward hacking, the predicted reward of the learned policy is high but the actual reward is low. The paper shows multiple lines of evidence that the observed reward hacking stems from spurious/non-causal correlations: Firstly, reward hacking disappears when non-causal features are removed from the input space. Secondly, the predicted reward is sensitive/salient to non-causal features that we know do not cause ground truth reward. Thirdly, after learning a policy, the state-action distribution differs from the training distribution, more than it does when non-causal features are not present. Thus, non-causal features exacerbate distribution shift, breaking spurious correlations even more. Then, the paper shows (a little bit of) evidence that causal confusion is exacerbated by noisy preferences, greater model scale, and unobservability of features that cause reward.

**Summary Of The Review:**

The paper addresses an important, interesting, and understudied problem by showing a variety of empirical evidence for spurious correlations and factors that exacerbate it in RL from preferences. To fully recommend acceptance it would need stronger evidence for its conclusions, or focus on conclusions that can be supported well.

EDIT: my concerns are sufficiently addressed now that I score the paper as marginally above acceptance.

---

> ### Author Response · Authors · 2022-11-16
> **Response to Reviewer hap7 (1)**
>
> Thank you very much for your thorough review. We've addressed your concerns and feedback in the latest revision of the paper (in blue text). Our responses to several of your comments are below:
>
> > The authors should discuss the relationship to goal misgeneralization (Langosco et al and Shah et al)
>
> Sounds good – we added the following to our related works section: Langosco et al. and Shah et al. study the RL setting with a known goal where “the features of the environment are correlated and predictive of the reward on the training distribution but not OOD.” This setting is similar to ours in that there is misidentification of the state features that are causal with respect to the reward (e.g. going to the end of a maze rather than collecting the coin that is the true reward). However, in contrast to our work, during training the reward signal is always present and always the ground-truth reward. Thus, we see their work as complementary to ours and have added it to our related work section. We show that when learning a reward function the learned reward can be misidentified, leading to poor RL performance. By contrast, Langosco et al. and Shah et al. show that even if the learned reward is perfect, the RL optimization can still fail due to spurious correlations during policy optimization.
>
> > The authors should also discuss to what extent the causal confusion problem in imitation learning is similar or different from the problem in preference learning.
>
> Thank you for the suggestion! We added a definition of the causal confusion problem in preference learning (Section 2) that can be used to compare it to the previously explored problem in imitation learning via behavioral cloning [1]. Notably, rather than discussing the causal confusion over actions (as previous work does), we discuss the causal confusion over the reward (which subsequently influences the actions).
>
> [1] De Haan, Pim, Dinesh Jayaraman, and Sergey Levine. "Causal confusion in imitation learning." Advances in Neural Information Processing Systems 32 (2019).
>
> > The second piece of evidence is that removing non-causal features improves performance. This is valid but only shown in one environment (Reacher). In the other environments, results are unclear or not visible (Fig 7).
>
> Table 5 shows additional KL divergences for the Feeding and Itch Scratching environments. We note here that the reason removing non-causal distractor features appears to not benefit for Feeding is that the main spurious correlation (discussed in Section 4) involves one of the causal features—namely, the difference between spoon position and target position. Thus, removing purely non-causal features fails to address this issue in Feeding. We also added a plot (Fig. 10d) where the effect of removing non-causal features is visible for Itch Scratching.
>
> > The third piece of evidence is that a saliency map shows that features are rewarded that are not causally related to ground truth reward. However, it must also be shown that these features are correlated with reward on the training data, otherwise no spurious correlation has been shown.
>
> Thanks for the suggestion! For each environment (each scenario discussed at the end of Section 4), we added saliency maps and correlation plots that support the claims about spurious correlations that certain features exhibit with the reward. These are located in Appx. A3.
>
> > Claim 2: a) preference noise worsens causal confusion and b) this becomes worse with more data. a) is not shown; the paper only seems to show that adding noise worsens task success, not causal confusion. b) is not given a convincing explanation, which means we need strong empirical evidence, ideally from more than two environments (Reacher and Feeding). … Why would trajectories become more similar when more samples are drawn but the distribution they are drawn from does not change?
>
> In Fig. 5a, we show that test accuracy stays high while policy performance plunges, which indicates causal confusion. In addition, we note that KL divergence between reward
> learning and reinforcement learning state distributions is greater when the reward is learned from stochastic user data, which supports the claim that adding noise worsens causal confusion. In the second paragraph of Section 5.3, we also clarify what we meant in b): for a fixed β and a fixed range of trajectory qualities (say, from rmin to rmax), increasing the number of trajectories increases the number of similar trajectories (the ‘density’ of trajectories in reward-space), which in turn increases the likelihood of drawing a pair of similar trajectories and mislabeling. That is, we increase data by increasing the number of trajectories while holding the range of trajectory qualities constant.

---

> > ### Author Response · Authors · 2022-11-16
> > **Response to Reviewer hap7 (2)**
> >
> > > Claim 3: greater model size worsens causal confusion. Figure 9 shows that this sometimes happens and some times the opposite happens. I don't think there is enough evidence here to make a claim that belongs in the abstract.
> >
> > We agree and have removed this from the abstract.
> >
> > > Section 4.5: The experiment simply shows that the learned policy fails. However, it does not show that this is due to causal confusion, reward hacking, and/or distribution shift.
> >
> > We added gradient saliency maps and KL divergence values to this section and show that the learned reward without ‘scratched‘ leads to a greater amount of distribution shift (Table 4).
> >
> > > Lastly, I was unsure what the main contributions are. Section 3 suggests that a key contribution is to show that the reason for reward hacking is causal confusion. This is also implied in the introduction: "we demonstrate the failure of preference-based reward learning to produce causal rewards". However, the abstract does not mention this contribution.
> >
> > Thank you for pointing this out. Yes this is a contribution and we have clarified this in the abstract. The last two paragraphs of the introduction also discuss our contributions.

---

> > ### Comment · Reviewer_hap7 · 2022-11-17
> > **Thank you for elaborating**
> >
> > Thank you for elaborating and for the extra results shown.
> >
> > Claim 1:
> >
> > - 2nd piece of evidence: The clarifications help. But they won't be visible to most readers. I'd strongly encourage having a single place in the main paper where you describe all pieces of evidence for this claim.
> >
> > - 3rd piece of evidence: This makes the evidence somewhat stronger. Again, it would help if evidence located in the appendix if actually described in the main text, not just referred to.
> >
> > Claim 2: a)
> > > In Fig. 5a, we show that test accuracy stays high while policy performance plunges, which indicates causal confusion.
> >
> > This indicates bad OOD generalization (where we're out-of-distribution because the policy optimization phase introduces new states). But I am missing evidence that bad OOD generalization must be due to spurious correlations.
> >
> > > In addition, we note that KL divergence between reward learning and reinforcement learning state distributions is greater when the reward is learned from stochastic user data, which supports the claim that adding noise worsens causal confusion.
> >
> > Again, this supports that noise increases distribution shift, but not necessarily that this leads to worse causal confusion (it's plausible but not directly shown).
> >
> > > increasing the number of trajectories increases the number of similar trajectories (the ‘density’ of trajectories in reward-space)
> >
> > It increases the number of similar trajectories but also the number of dissimilar trajectories. So I'm still confused why it should become more likely to randomly draw similar trajectories. (I also do not follow why fixing the range of trajectory qualities relates to this argument.)
> >
> > Additionally, even if collecting more preferences leads to a higher share of mislabeled preferences, it remains unclear why this would worsen causal confusion.
> >
> > My current impression is that the experiments about noise could be removed, and the paper would still have enough to say, and have more space to support it.
> >
> > Overall, the soundness of some claims is too unclear to me to increase my score, although you have supported claim 1 better than before. Although the paper supports interesting and practically important hypotheses, and I'm definitely close to recommending it, I think it would be stronger with more comprehensive and clearly presented evidence for precise claims.

---

> > > ### Author Response · Authors · 2022-11-18
> > > **Follow-up to Reviewer hap7**
> > >
> > > Thanks again for the detailed feedback! We've uploaded a new revision with changes described below:
> > >
> > > > But they won't be visible to most readers. I'd strongly encourage having a single place in the main paper where you describe all pieces of evidence for this claim. … Again, it would help if evidence located in the appendix if actually described in the main text, not just referred to.
> > >
> > > Got it, thanks for the suggestion. We added the clarifications about removing distractor features in other environments to the end of Section 5.1. We also made sure to describe the correlation plots in Section 4 as we mention them, rather than simply referencing the appendix.
> > >
> > > > ​​This indicates bad OOD generalization (where we're out-of-distribution because the policy optimization phase introduces new states). But I am missing evidence that bad OOD generalization must be due to spurious correlations.
> > >
> > > We agree, and feel that ‘causal confusion’ is an overloaded (and thus ambiguous) term to use in this case. We have edited the paper to introduce the term ‘causal reward misidentification’ instead of causal confusion, and defined it exactly as good ID performance but bad OOD performance. Our stance is that if the learned model fails OOD (but does well ID), it has failed to identify some causal aspect of the reward — if it identified everything perfectly, then it would also generalize. If we think of the reward as features and their combinations, then a misidentification can occur either because the network constructs another feature that correlates with a causal one ID, or because the network combines features in a way that correlates with the causal combination ID. In that sense, any misgeneralization is due to some causal signal not being identified correctly, thus the term ‘causal reward misidentification’. We think the (updated) claims are well supported, since results consistently show good ID performance and bad OOD performance. The reviewer is correct that  we can’t always trace this back to a particular spurious feature we can identify that the model picks up on. Nonetheless, it feels pretty important to know that reward learning can fail this way.
> > >
> > > > It increases the number of similar trajectories but also the number of dissimilar trajectories. So I'm still confused why it should become more likely to randomly draw similar trajectories.
> > >
> > > Thanks for pointing this out. We agree that we can only claim that we have more incorrect preferences AND more correct ones. We have removed this explanation from the paper in the most updated revision.

---

> > > > ### Author Response · Authors · 2022-11-18
> > > > **Follow-up to Reviewer hap7 (cont.'d)**
> > > >
> > > > We also wanted to follow-up to reiterate that, based on our understanding and our definition of causal reward misidentification, Table 1 provides strong evidence for reward misidentification because the reward model’s test accuracy is high, the performance of the resulting policy is low under the true reward, and the learned reward prefers the resulting policy over an expert RL policy. This shows that it's not an RL problem, it's a problem with the incentives provided to RL, namely the misidentified reward function.
> > > >
> > > > We wanted to ask whether this has clarified your questions and strengthened the paper, and whether you are willing to increase your score. If not, are there any specific experiments that we can run or areas we can address that could help you be willing to increase your score?

---

> ### Author Response · Authors · 2022-11-17
> **Re: Response to Reviewer hap7**
>
> Dear reviewer hap7,
>
> We are grateful for your time and apologize for the intrusion, but as tomorrow is the last day we can update the draft, we are wondering if our response makes sense and if there are any other changes you’d like to see. Thank you!

---

### Official Review · Reviewer_DXr6 · 2022-10-27

**Confidence:** 3
**Correctness:** 3
**Technical Novelty And Significance:** 2
**Empirical Novelty And Significance:** 3
**Recommendation:** 6

**Clarity, Quality, Novelty And Reproducibility:**

**Clarity**

The presentation is generally clear and concise. Sometimes a lot of flipping between text and appendix is required. I feel like the plots could be organized more coherently to be on the pages where they're mentioned. The plots themselves are simple, easy to understand and convey the desired message well.

**Quality**

Quality of writing is very high. It has hardly any typos/grammatical errors.

**Novelty**

This work's main novelty lies in the application of existing methods for analysis to an interesting problem setting. Apart from this, the authors provide a method for approximating the KL divergence between two data distributions. Novelty is thus a weakness of the paper.

**Reproducibility**

The authors state that they will release source code and datasets. Reward functions are not stated in the paper but the environments used by the authors are open source. No hyperparameters for the RL agents are given. With a release of those I think the work can be fully reproduced.

**Strength And Weaknesses:**

**Strengths**

- Assistive robotics is a very interesting and relevant problem domain.
- The analyzed failure cases are of high practical relevancy.
- The empirical analysis is easy to follow.
- Videos on the companion website qualitatively illustrate the failure modes of trained RL agents very well.
- I appreciated the authors making a concerted effort to convey their findings graphically.

**Weaknesses**
My main concerns regarding the paper are:

- The paper has limited novelty. While the setting is new and interesting, the methods used by the authors to measure causal confusion are not novel, except for a method to approximate the KL divergence between two distributions of state-action pairs.
- No proposed solution method. The paper reads like a very nice and thorough exposition of an important problem but then it "abruptly" ends. Once the authors introduced their method to measure the KL divergence between state-action distributions, I was immediately thinking of whether this could be used to penalize learned rewards and thereby encouraging agents to stay within the "known" region of the state-action space. Even negative results on "straightforward" (devil's in the details of course) remedies like this would be nice.

More minor weaknesses:

- Noise evaluations could be more detailed. I agree with the authors that user noise is inevitable, but in my opinion the setting they evaluate ($\epsilon$-greedy action noise) is rather uninteresting. "*B-Pref: Benchmarking Preference-Based Reinforcement Learning*" (Lee et al.) introduces several types of noise based on known cognitive biases. It would be interesting to see how such structured noise affects performance and I also think evaluating it would mesh very well with the practical relevancy of the author's work.
- It would have been nice to have an explanation of the noise procedure in the main paper as I think it's relevant to understanding.
- The authors evaluate against the ground-truth reward functions very often. It's frustrating that these are neither stated in the main paper nor in the appendix.
- Hyperparameters for the SAC and PPO agents where missing.
- Evaluation methods are not used consistently for all settings. There are no saliency maps and EPIC distances for itch scratching. They would help paint a more complete picture even though I presume they are omitted because the agent fails to learn in most settings anyway.
- In A.4: "*With the trained model, we calculate $D_{\text{KL}} (p \parallel q)$ by taking the negative mean return/logit of all
reward learning observation-action pairs*". I assume it's the mean return according to the reward model? Would still be nice to have it explicitly stated here.

**Summary Of The Paper:**

This work analyzes the occurrence of spurious correlations in learning reward functions from preferences in the commonly used Bradley-Terry model. The authors show that picking up such correlations can have a drastic effect on the performance of an RL agent that is trained using the learned reward. In their empirical analysis, the authors examine the effect of distractor features, noise, partial observability and model capacity. The paper quantifies how these factors affect the trained RL agent on the basis of several different metrics.

**Summary Of The Review:**

Overall, I'm torn on this paper. On one hand, the authors analyze a relevant problem in a very interesting and relevant problem domain. The writing is clear and easy to understand and comes with detailed empirical analysis that follows a coherent structure.
On the other hand, this feels like reading only the first half of the paper: After the problem has been clearly identified, a (partial) solution is missing. Of course that's not necessary for each paper, but without a solution method the limited novelty and in points not 100% clear presentation weigh more heavily.
For me, the lacking novelty currently outweighs the intriguing setting and therefore I'm voting for weak reject. I'm looking forward to an improved version of the paper (even just addressing some of the weaknesses) because there's obvious potential here.

---

> ### Author Response · Authors · 2022-11-16
> **Response to Reviewer DXr6**
>
> Thank you so much for your detailed suggestions! We found them to be very helpful in writing our revision (especially the suggestions about a proposed solution and a more detailed analysis of user noise using B-Pref). Detailed responses are below, and changes to the paper are written in blue text.
>
> > No proposed solution method. … Once the authors introduced their method to measure the KL divergence between state-action distributions, I was immediately thinking of whether this could be used to penalize learned rewards and thereby encouraging agents to stay within the "known" region of the state-action space. Even negative results on "straightforward" (devil's in the details of course) remedies like this would be nice.
>
> Thank you for the suggestion. We agree that a discussion of potential solutions would strengthen the paper and have added this discussion as a new paragraph in the conclusion. We highlight ideas involving penalizing divergence from the training distribution, using human input to remove non-causal features, and using human input to better learn high-level causal features such as the “itch” feature described in our paper. Following your suggestion, we explore a solution that penalizes the KL divergence in Appx. A12. We find that, unfortunately, the RL agent is again able to hack the reward---this time, the performance and causal confusion are even worse because of the additional degree of freedom in the reward function afforded by the discriminator penalty. We’ve also performed a preliminary exploration of the use of active learning and querying for preferences online, located in Appx. A13.
>
> > Noise evaluations could be more detailed. … "B-Pref: Benchmarking Preference-Based Reinforcement Learning" (Lee et al.) introduces several types of noise based on known cognitive biases. … It would have been nice to have an explanation of the noise procedure in the main paper as I think it's relevant to understanding.
>
> Thanks for the suggestion! We follow the framework suggested by B-Pref (Section 5.3) to explore four types of user noise (varied independently of each other): STOCHASTIC, where the user is rational with $\beta=1$; MYOPIC, where earlier rewards are discounted with a $\gamma=0.99$; SKIP, where the teacher/preference-giver skips a pair of trajectories if both of have rewards below a certain threshold; and MISTAKES, where the preference label is randomly flipped with probability $\epsilon=0.1$.
>
> > The authors evaluate against the ground-truth reward functions very often. It's frustrating that these are neither stated in the main paper nor in the appendix. Hyperparameters for the SAC and PPO agents where missing.
>
> Thank you for pointing this out. We added descriptions of the ground-truth rewards in Appendix A5 and hyperparameter values in Appendix A1.
>
> > Evaluation methods are not used consistently for all settings. There are no saliency maps and EPIC distances for itch scratching.
>
> We went through and added these to the main paper or appendix, as appropriate. Notably, we added saliency maps for Itch Scratching in Fig.9 and KL divergences for Itch Scratching in Table 4.
>
> > In A.4: "With the trained model, we calculate D_kl by taking the negative mean return/logit of all reward learning observation-action pairs". I assume it's the mean return according to the reward model?
>
> Actually, we calculate the KL divergence “by taking the **discriminator's** negative mean return/logit.” We make sure to clarify this in Appx. A6.

---

> > ### Comment · Reviewer_DXr6 · 2022-11-19
> > **Thank you for addressing my points**
> >
> > While I believe that my main two critic points are still somewhat valid, the authors addressed many of the smaller issues I had with the paper, thereby improving its clarity. I think the experiments with actual biases are valuable for a work with practical implications such as this one  - expanding them to the other environments would be nice!
> >
> > Thank you for adding the KL-divergence experiments. A negative result on this straightforward solution in my eyes serves an important role by a) highlighting the severity of the problem and b) by showing that primitively forcing the policy to adhere to the reward model's training data is not enough, which provides additional insight into the reward-hacking issue. I commend the authors for adding it.
> >
> > In light of the improvements made by the authors I will raise my score to 6 (weak accept). Good luck!

---

> ### Author Response · Authors · 2022-11-17
> **Re: Response to Reviewer DXr6**
>
> Dear reviewer DXr6,
>
> We are grateful for your time and apologize for the intrusion, but as tomorrow is the last day we can update the draft, we are wondering if our response makes sense and if there are any other changes you’d like to see. Thank you!

---

### Official Review · Reviewer_DzdV · 2022-10-27

**Confidence:** 4
**Correctness:** 2
**Technical Novelty And Significance:** 2
**Empirical Novelty And Significance:** 1
**Recommendation:** 5

**Clarity, Quality, Novelty And Reproducibility:**

The paper is well-written and clear.

However, the novelty of the work is limited as well as its scope, which lessens the contributions.

In addition, some claims in Section 4 seem to not be fully supported. For example, in Section 4.1., "in incentivizing the Reacher robot to spin fast, it leads the RL optimization toward states that were not seen during reward learning" ; might an exploration issue be the cause of the poor performance? Another example in Section 4.3., where it is stated that adding "preference data when noise is present actually has a negative effect"; might this be a specific phenomenon that happens with the selected noise model? Does this happens under additional models? We would expect noise to be averaged over different samples so that adding more data would increase performance.

A few small typos need to be corrected: e.g., "to have a access" in Sec. 1 -> an access , "Note that later sections, we observe" -> in late sections, in Sec. 3

**Strength And Weaknesses:**

The paper is well-written and the ideas are exposed clearly.

However, the scope of this study is very limited. The experimental setup only includes continuous robot learning benchmarks, with three similar tasks, and is not extended to additional, potentially very different, environments. Besides, a specific model is used for the preferences. It might also be interested to compare the results here with alternative preference-based reinforcement learning approach, including additional value-based approaches and relation-based techniques (see for example the review: "A Survey of Preference-Based Reinforcement Learning Methods", Christian Wirth et al.). Therefore, it is not clear if the conclusions reached in this study may be extended beyond this narrow scope.

Another limitation of this work is the use of only synthetic preferences, computed from the real reward using a noise model. Would the same conclusions hold with real human raters preferences?


**Summary Of The Paper:**

This work studies causal confusion in the context of reward learning from preferences. It is shown that rewards learning from preferences achieve poor performance in terms of learning a policy in three robot learning benchmarks. Several factors for causal confusion are identified, including distractor features and noise in stated preferences.

**Summary Of The Review:**

This work considers an interesting idea. However, the narrowness of this scope of this study as well as the lack of full support for certain claims limits the contribution of this work. It seems that it is not yet ready for publication and a larger scope would improve considerably its contribution.

---

> ### Author Response · Authors · 2022-11-15
> **Response to Reviewer DzdV**
>
> Thank you so much for your detailed feedback! Below are responses to some of the weaknesses that you pointed out / suggestions that you had. (We've added revisions to the paper in blue text for ease of review.)
>
> > It might also be interested to compare the results here with alternative preference-based reinforcement learning approaches mentioned in Wirth et al.
>
> As noted by Wirth et al., Value-based approaches are difficult to scale since they require access to the transition dynamics and taking an argmax over actions which is difficult in continuous action spaces. Thus, to the best of our knowledge, they have also only been applied to preferences over individual states, whereas we look at trajectory preferences which we believe to be more intuitive given that most recent works that study user-provided preferences use trajectory preference labeling [1,2,3,4]. Finally, as noted by Wirth et al. these approaches do not directly lend themselves to RL solutions in the same way that reward-based utility functions do. Wirth et al. note that reward-based utility functions are independent of the dynamics and thus, in theory, should be simpler to learn and to generalize. They also note that “reward-based utilities seem to be appropriate for domains without well defined policy spaces, because they allow the application of efficient value-based reinforcement learning methods.” Thus, while we agree that future work should study different classes of preference-based RL methods (e.g. value-based and relation-based), we believe that the reward-based utility approach is the most common approach in current practice and also the one noted by Wirth et al. as the most general and scalable.
>
> [1] Christiano, Paul F., et al. "Deep reinforcement learning from human preferences." Advances in neural information processing systems 30 (2017).
>
> [2] Ibarz, Borja, et al. "Reward learning from human preferences and demonstrations in atari." Advances in neural information processing systems 31 (2018).
>
> [3] Erdem, B., et al. "Asking Easy Questions: A User-Friendly Approach to Active Reward Learning." Conference on Robot Learning. PMLR, 2020.
>
> [4] Lee, Kimin, Laura Smith, and Pieter Abbeel. "PEBBLE: Feedback-Efficient Interactive Reinforcement Learning via Relabeling Experience and Unsupervised Pre-training." International Conference on Machine Learning. 2021.
>
> > Another limitation of this work is the use of only synthetic preferences, computed from the real reward using a noise model. Would the same conclusions hold with real human raters preferences?
>
> Great question. A key characteristic of data from real humans is a variety of noise and biases. Inspired by this, we have added new results studying different forms of noise. We follow the B-Pref [1] framework to explore the various effects of human user bias/noise (myopic users, stochastic users, users that make random mistakes, etc.). We added these new results to Section 5.3 of our paper.
>
> [1] Lee, K., et al. "B-Pref: Benchmarking Preference-Based Reinforcement Learning." Neural Information Processing Systems (NeurIPS) (2021).
>
>
> > Might an exploration issue be the cause of the poor performance?
>
> In Table 1, we find that the learned reward does prefer the policy trained using the learned reward over the policy obtained by optimizing the true reward. This shows that the poor performance is not due to an issue in RL exploration. We added a comment addressing this in Section 4.
>
> > We would expect noise to be averaged over different samples so that adding more data would increase performance.
>
> To clarify, when increasing our dataset size, we increase the number of trajectories while holding the rationality constant $\beta$ and the range of trajectory qualities constant. Thus, for a fixed $\beta$ and a fixed range of trajectory qualities, increasing the number of trajectories increases the number of *similar* trajectories (the `density' of trajectories in reward-space), which in turn increases the likelihood of drawing a pair of similar trajectories and mislabeling.

---

> ### Author Response · Authors · 2022-11-18
> **Re: Response to Reviewer DzdV**
>
> We wanted to provide some updates we've made since our previous response:
>
> > The experimental setup only includes continuous robot learning benchmarks, with three similar tasks, and is not extended to additional, potentially very different, environments.
>
> In Appendix A3 of the most recent revision, we’ve added results from the Lunar Lander environment [1], which involves discrete actions and is a popular control task that is quite different from the assistive gym and reacher tasks (which all involve reaching toward some sort of target). We found that the results are in line with our findings.
>
> We wanted to ask whether this has clarified your questions and strengthened the paper—if so, would you be willing to increase your score in light of these updates? If not, are there any specific experiments that we can run or areas we can address that could help you increase your score?
>
> [1] Greg Brockman, Vicki Cheung, Ludwig Pettersson, Jonas Schneider, John Schulman, Jie Tang, and Wojciech Zaremba. Openai gym. arXiv preprint arXiv:1606.01540, 2016

---

> ### Comment · Reviewer_DzdV · 2022-11-28
> **Response to the authors**
>
> Dear Authors,
>
> Thank you for your response as well as the updates performed. I retain my opinion that the paper's contribution is limited. However, following the additions made to the manuscript, I've raised my score to 5 (weak reject).

---

### Author Response · Authors · 2022-11-18
**Response to all reviewers (1)**

Thank you to all the reviewers for your time and the extensive, detailed feedback you have provided. We have made significant updates to our paper in response to your feedback, and we feel that it is much stronger than before, thanks to you all. We summarize the changes below:

- We recognize that ‘causal confusion’ as a term is overloaded and thus ambiguous to use without context. As such, we added a section (Section 2, Causal Reward Misidentification) describing and defining the exact problem we study in this paper—namely, when a learned reward achieves low test error on in-distribution data but results in poor policy performance due to RL-induced distribution shift—which we call “causal reward misidentification”. In this section, we also include a causal graph in Fig. 1 that describes the observable and unobservable variables in the preference-based reward learning setting and elucidates the potential confounders and sources of causal reward misidentification. The idea to include a formal definition was first brought up by Reviewer Ek1w but also addresses Reviewer hap7’s concern about comparing our work with the equivalent problem in imitation learning via behavioral cloning. Reviewer hap7 is correct that we can’t always trace causal reward misidentification back to a particular spurious feature or training data correlation—in general, this is still an open question for deep learning methods. Nonetheless, we believe it is very important to know that reward learning can fail this way and we believe our findings will be interesting for the research community and lead to many interesting areas of future work.

- We took several steps to strengthen our claims (and remove weak ones).
  - We clarified that poor policy performance is not due to an issue in RL exploration because the learned reward prefers the policy trained using the learned reward over the policy obtained by optimizing the true reward (Section 4, Table 1) (Reviewer DzdV)—hence, the learned reward fundamentally misidentifies at least a part of the causal structure.
  - Further, we included saliency maps for the misidentified rewards in all three environments discussed in the body of Section 4 to solidify our claim that the reward network has weighted/learned a particular feature incorrectly. On top of this, we provided correlation plots of the identified spuriously weighted feature and the true reward in order to show how, in this case, the faulty reward is due to a spurious correlation in the training data (Reviewer hap7).
  - As recommended by Reviewer DzdV, we also added new experiments on a different domain in Appx. A3. We study the Lunar Lander environment, a discrete classic control task, and demonstrate that causal reward misidentification is still a problem when learning from offline preference data.
  - We were more thorough about including results for other environments and clarifying any discrepancies in Section 5.1 (Reviewer hap7).

- We took several steps to enhance clarity and reproducibility.
  - We described the ground truth reward functions in Appx. A5 and list the PPO/SAC hyperparameters in Appx. A1 (Reviewer DXr6).
  - We added an algorithm outline of our preference-based reward learning procedure in Algorithm 1 to clarify the sequence of data collection, reward learning, and policy optimization (Reviewer VP1L).

- We added a section to our Related Work about Goal Misgeneralization, which shows failures that occur due to when the researchers induce a test-time distribution shift and evaluate a pretrained RL policy that was trained with a known reward function (Reviewer hap7). This work is complementary to ours and shows that even if we perfectly recover the right reward, if the environment changes, then a learned policy may still fail. By contrast, our work studies the case of an unobserved reward that must be learned and studies and analyzes factors that lead to having a misidentified reward function. In our setting, it is the misidentified reward that causes RL itself to induce distribution shift, leading to failures during training, despite no changes in the environment.

---

> ### Author Response · Authors · 2022-11-18
> **Response to all reviewers (2)**
>
> - We added a new set of experiments exploring different types of (more realistic) user noise, inspired by the B-Pref benchmark [1] (Reviewer DXr6). This provides an insight into what would happen if the data were provided by real human raters (Reviewers DzdV and DXr6), as well as commonalities/differences in the effects of different noise models (Reviewer DzdV). We also clarify and formalize the different types of noise we explore using Equation 3 (Reviewer DXr6). We find across multiple noise models that causal reward misidentification is exacerbated by noise. We verify causal reward misidentification by showing in Fig. 5a that reward network test accuracy is not predictive of resulting policy behavior; test accuracy stays high while policy performance drops, which matches our definition of causal reward misidentification. We also calculated KL divergence estimates (Table 9) to verify that these misidentified rewards resulted in greater distribution shift during RL.
> [1] Lee, K., et al. "B-Pref: Benchmarking Preference-Based Reinforcement Learning." Neural Information Processing Systems (NeurIPS) (2021).
>
> - As suggested by Reviewers DXr6 and Ek1w, we added a discussion in the Conclusion of promising ideas for future work that could address causal reward misidentification. These ideas are inspired by the findings in our paper. Furthermore, we added and discuss the following new experiments studying some straightforward approaches:
>   - Our results show that causally confused rewards result in greater distribution shifts during policy optimization. In Appx. A12, we examine the effect of adding a direct penalty on the KL divergence into the cost function during RL. We find that, unfortunately, the RL agent is again able to hack the reward---this time, the performance and causal reward misidentification are even worse because of the additional degree of freedom in the reward function afforded by the discriminator penalty. However, we hypothesize that there may be other less obvious ways to successfully incorporate such a penalty (Reviewer DXr6).
>   - We recognize that active and iterative methods for data acquisition is also a widely popular solution---as such, we provide a preliminary exploration in Appx. A13 and find that causal reward misidentification still appears to be present (Reviewer VP1L).
>
> We thank the reviewers for their invaluable comments that inspired these revisions to our paper—we sincerely believe that we have addressed the main concerns of the reviewers and that our paper has significantly improved. In light of this, we respectfully request that the reviewers increase their scores if we have addressed their main concerns. If not, are there any specific questions we can address that would help you increase your score?

---

> ### Comment · Reviewer_hap7 · 2022-11-30
> **Small comment on terminology**
>
> Small comment:
>
> Correct me if I'm wrong but 'causal reward misidentification' could simply be called 'reward misidentification' (or 'reward misgeneralization' following Langosco et al). Nothing about the definition seems to imply that the misidentification is related to causality. The definition is simply that the reward model predicts well in distribution and badly out of distribution. Even if the data has no causal confusion / spurious correlation at all, a model can still generalize badly out of distribution. Thus it seems that the word causal should be dropped. This would reduce confusion for me.

---

> > ### Author Response · Authors · 2022-11-30
> > **Response about Terminology**
> >
> > This is a good point. In our minds, there is one true (causal) hypothesis, consisting of the correct variables and the correct way to combine those variables into a reward. But this discussion points out that causality can be thought of to be more specific (eg., a specific variable that affects the reward correlates with a different one), which is what prompted the comments on causal confusion to begin with. Referring to it as reward misidentification very clearly points to “this is not the correct reward function”, so we will take that suggestion and revise the paper.

---

> ### Comment · Reviewer_hap7 · 2022-12-12
> **Final update**
>
> (Copied from the private discussion)
>
> I'm happy for this to be published conditional on some easy changes that should be reflected in the abstract and intro. I think the paper needs to tone down or remove some claims, but I also think that the main message is useful and important: that reward functions are easily misidentified (sometimes due to spurious correlations) and this only shows when the RL-trained policy drives the state off the training distribution, leading to bad performance.
>
> Specifically, I encourage the authors to
> 1) consider removing claims about noise because the authors have acknowledged that their rationale for these was incorrect. (Correct me if I misinterpreted)
> 2) tone down claims that reward misidentification is due spurious correlations / causal confusion since the evidence is limited. In particular, where the paper talks about 'causal confusion / spurious correlations', it should often refer to reward misidentification instead.
> 3) remove the claim that greater model size worsens causal confusion. Figure 9 (original submission) shows that this sometimes happens and some times the opposite happens. I don't think there is enough evidence here to make a claim that belongs in the abstract. The authors have not addressed this concern yet.

---

> > ### Author Response · Authors · 2022-12-13
> > **Re: Final update**
> >
> > Thank you for your constructive and helpful feedback!
> >
> > > consider removing claims about noise because the authors have acknowledged that their rationale for these was incorrect. (Correct me if I misinterpreted)
> >
> > Yes—we have removed our (faulty) explanation for why increasing the noise level increases the likelihood of drawing similar trajectories. We choose to keep the section on Noise in Stated Preferences due to the concerns brought up by Reviewers DzdV and DXr6 (explained further in our public comment, “Response to all reviewers (2)”).
> >
> > > tone down claims that reward misidentification is due spurious correlations / causal confusion since the evidence is limited. In particular, where the paper talks about 'causal confusion / spurious correlations', it should often refer to reward misidentification instead.
> >
> > Sounds good. We have edited the paper to use “reward misidentification” rather than 'causal confusion / spurious correlations’—since the phase for submitting revisions has closed, this change will be reflected in the camera-ready version.
> >
> > > remove the claim that greater model size worsens causal confusion. Figure 9 (original submission) shows that this sometimes happens and some times the opposite happens. I don't think there is enough evidence here to make a claim that belongs in the abstract. The authors have not addressed this concern yet.
> >
> > We have already addressed this in prior revisions (in the **PDF** of the paper). Unfortunately, we were not able to edit the version of the abstract that is displayed on OpenReview (the platform did not let us edit it), but we wanted to assure the reviewers that we have removed the claim that “larger model capacity can exacerbate causal confusion” in the **PDF revision**, and will update the abstract displayed on OpenReview as soon as we are able to.
> >
> > We very much appreciate the recommendations to improve our paper, agree that they are easy changes to make, and will make sure to address them in the camera-ready version, if they have not been addressed already. In light of this, we hope you will consider raising your score.

---

### Author Response · Authors · 2022-11-30
**Regarding the Novelty and Contribution of our Paper**

Thank you to all the reviewers for your responses thus far! Recent reviewer comments have mentioned lingering concerns about the **novelty** and **contribution** of our paper. We wanted to respond to these concerns directly in this comment.

With respect to novelty (Reviewer DXr6), the concern was that there is no algorithmic novelty (other than perhaps the way we approximate divergence). We agree, but we feel strongly that this is not the only way for a paper to have novelty. Our paper is, to our knowledge, the first systematic study of reward misidentification (good test set accuracy, but poor performance when optimized). This is a big issue. Prior work has only touched on this topic anecdotally, without examining the problem in more detail. We provide the first concrete definition of reward misidentification in preference-based reward learning and provide evidence for its existence, how it is exacerbated, and the consequences that result. Further, the complexity of our task environments is much higher than that of prior work dealing with causal confusion in imitation learning—the state- and action-spaces for the Feeding and Itch-Scratching tasks (from Assistive Gym) are quite a bit larger than those of MountainCar, LunarLander, or the MuJoCo environments (used in prior work). The true rewards on which the preferences are based are also significantly more complex in the Assistive Gym environments.

With respect to the concern about the paper’s contribution (Reviewer DzdV), our paper presents an analysis, which is a type of study that we believe is very important but often overlooked in ML. The findings from our analysis open up lots of interesting areas for future work: how to deal with partial observability of causal features, how to penalize unnecessary distribution shift during policy training, what the best type of active interventions to disentangle causality are, and how to use human input most efficiently to avoid reward misidentification, to name a few.

Finally, Reviewer DzdV expressed concerns regarding the generalizability of the results. We are reassured by others who discussed the topic of reward misidentification anecdotally that there is a larger underlying phenomenon, but the details of the effects will indeed need to be reproduced in other areas—this will be an issue with any analysis paper. However, as mentioned above we believe our environments cover complexities previously unexplored and that our results and analysis will be of interest to many in the ML community, especially given the growing interest in human-AI alignment, preference and reward learning, and causality.

---

### Decision · Program_Chairs · 2023-01-20

**Decision:**

Accept: poster

**Justification For Why Not Higher Score:**

Several issues which should be addressed in the camera ready version of the paper. Analysis would have to be extended to warrant a higher score.

**Justification For Why Not Lower Score:**

The paper contains material interesting to the community.

**Metareview: Summary, Strengths And Weaknesses:**

This paper addresses the problem of causal confusion in preference-based reward learning. In particular, the setting in which low test error is achieved with the learned reward function but out-of-distribution-generalization fails, is analysed on several benchmark datasets. The authors identify several factors that can exacerbate causal confusion and demonstrate their impact in experiments.

Strengths of the paper:
* The paper is mainly well-written and most ideas and thoughts are clearly expressed.
* The considered problem is important and the study goes deeper than existing studies.
* The analyzed failure cases are of high practical relevancy.

Weaknesses of the paper:
* Limited number of experiments, only on the robotics domain.
* Only a single preference model is considered; the used noise model is of limited interest.
* Uses only synthetic preferences, computed from the real reward using a noise model.
* There is no conclusion in the sense of providing a method or clear directions for resolving the identified issues (if this is possible).


----

Discussion and recommendation:
The paper received mixed scores and, in most cases, a discussion between reviewers and authors took place. The authors improved their paper already addressing several of the reviewers' comments. In a final call, we came to the consensus that despite some shortcomings, the paper contains quite some material relevant to the community working on reward-identification and causal confusion. Hence I am recommending acceptance of the paper. Nevertheless, the authors are strongly encouraged to update their paper in line with the reviewers comments. In particular, regarding the following the claims (copied and adjusted from the corresponding reviewer's comments):

* Remove/clarify claims about noise that the authors have to be incorrect
* Tone down claims that reward misidentification is due spurious correlations / causal confusion since the evidence is limited. In particular, where the paper talks about 'causal confusion / spurious correlations', it should often refer to reward misidentification instead.
* Remove the claim that greater model size worsens causal confusion. Figure 9 (original submission) shows that this sometimes happens and some times the opposite happens. There's not sufficient evidence to clearly support that claim.

But also the other points raised by the reviewers are important and should be incorporated in the camera ready submission.



**Note From Pc:**

if the above contains the word "oral" or "spotlight" please see: "oral" presentation means -> notable-top-5% and "spotlight" means -> notable-top-25%. As stated in our emails, we are disassociating presentation type from AC recommendations